# Multi-omics analysis of parthanatos related molecular subgroup and prognostic model development in stomach adenocarcinoma

Xiangxin Wu[1‡], Lianming Cai[1‡], Yanping Liu[1‡], Bowen Wang[1], Tianyi Xia[2*], Zhenhua Liu[1*]

**1** Department of Abdominal Surgery, Ganzhou Cancer Hospital, Ganzhou, China, **2** Department of Colorectal Surgery, Harbin Medical University Cancer Hospital, Harbin Medical University, Harbin, China

‡ These authors have contributed equally to this work and share first authorship
* xiatianyi@hrbmu.edu.cn (TX); lzh225898@163.com (ZL)

## Abstract

Stomach adenocarcinoma (STAD), the most prevalent histological subtype of gastric cancer, exhibits high heterogeneity and poor prognosis, posing significant therapeutic challenges. Parthanatos, a distinct form of regulated cell death mediated by poly (ADP-ribose) polymerase-1 (PARP-1), has been implicated in tumor biology and therapeutic resistance; however, the role of parthanatos-associated genes (PRGs) in STAD remains largely unexplored. In this study, we performed a comprehensive multi-omics analysis integrating transcriptomic, genomic, and clinical data from public databases to delineate the molecular landscape of PRGs in STAD. Unsupervised clustering revealed distinct PRG-related molecular subtypes with significant differences in clinical outcomes, immune infiltration profiles, and biological pathway activation. Based on machine learning algorithms, we established and validated a novel PRG-based prognostic signature, which demonstrated robust predictive performance. Moreover, single-cell RNA sequencing and *in vitro* functional assays were conducted to explore cellular heterogeneity and gene function. Notably, *in vitro* experiments, including western blot, colony formation, CCK-8, and Transwell assays, confirmed that one key PRG, COL8A1, promotes STAD cell proliferation and migration. Collectively, our findings highlight the clinical and biological significance of PRGs in STAD, offering novel biomarkers and potential therapeutic targets for STAD precision treatment.

## Introduction

Stomach adenocarcinoma (STAD), the predominant histological subtype of gastric cancer, arises from the glandular epithelial cells of the stomach [1]. It is characterized by aggressive progression, late diagnosis, and marked molecular heterogeneity, resulting in a poor prognosis, with a 5-year survival rate of less than 30% in advanced

**Data availability statement:** All relevant data are within the paper and its Supporting Information files.

**Funding:** This project is supported by Project of Heilongjiang Provincial Health Commission (No. 20230404080327). The funders had no role in study design, data collection and analysis, decision to publish, or preparation of the manuscript.

**Competing interests:** The authors have declared that no competing interests exist.

cases [2,3]. Current treatment strategies include surgical resection, chemotherapy, targeted therapies, and immune checkpoint inhibitors [4]. Tumor heterogeneity leads to inconsistent treatment responses, posing persistent clinical challenges [5–7]. Additionally, biomarkers for patient selection remain insufficiently validated and acquired drug resistance often limits long-term efficacy [4,8–10]. Therefore, further research on the mechanisms of STAD is necessary.

Different from necroptosis, pyroptosis or ferroptosis, parthanatos is a recently recognized form of regulated cell death [11]. This cell death pathway is primarily triggered by DNA damage-induced activation of poly (ADP-ribose) polymerase-1 (PARP-1), a key enzyme involved in DNA repair [12]. Once activated, PARP-1 synthesizes Poly (ADP-ribose) (PAR) polymers, which translocate to the mitochondria and trigger apoptosis inducing factor (AIF) release [13]. AIF, upon entering the cytoplasm, interacts with macrophage migration inhibitory factor (MIF), which then translocates to the nucleus to mediate DNA fragmentation through its nuclease activity [14]. Key features of parthanatos include caspase independence, mitochondrial membrane depolarization, secondary reactive oxygen species (ROS) generation, dependence on calcium signaling, and resistance to the cytoprotective effects of Bcl-2 [15–18]. Through the above mechanisms, parthanatos participates in maintaining the balance between DNA repair and cell death [19].

Although the precise mechanisms remain unclear, emerging evidence suggests that parthanatos plays a critical role in tumorigenesis [20]. The histone demethylase KDM6B has been shown to enhance DNA damage by inhibiting DNA repair and checkpoint response mechanisms, leading to excessive PARP-1 activation and subsequent parthanatos [21]. The crosstalk between parthanatos and key necroptosis-related proteins has been implicated in tumor immune evasion in hepatocellular carcinoma, highlighting the role of parthanatos in the tumor-associated immune microenvironment [22]. In addition, tumor suppression induced by targeting the AKT pathway has been linked to parthanatos activation [23]. Evidence also indicates that the combination of apoptosis-inducing agents with PARP inhibitors can synergistically promote cell death [24]. In leukemia patients, PARP-1 serves as a biomarker for the sensitivity of cytarabine and idarubicin treatment regimens due to its role in parthanatos induction [25]. Therefore, parthanatos holds potential for overcoming treatment resistance and enhancing cancer therapy through a combined strategy involving both parthanatos and apoptosis [24]. Despite its emerging role in various cancer types, in-depth studies on parthanatos in STAD remain lacking.

Despite increasing interest in parthanatos in cancer biology, the expression landscape and clinical relevance of parthanatos-associated genes (PRGs) in STAD remain poorly characterized. Advances in multi-omics technologies have made further research possible [26–29]. In this study, we systematically characterized the expression patterns, genomic alterations, and functional roles of PRG signatures in STAD. By integrating multi-omics analyses, machine learning modeling, single-cell RNA sequencing and *in vitro* validation, we identified distinct PRG-related molecular subtypes, constructed a robust prognostic scoring model, and revealed COL8A1 as a key PRG that may contribute to tumor progression and immune evasion. Collectively,

our findings offer novel insights into the biological and clinical relevance of PRGs in STAD, highlighting potential biomarkers and therapeutic targets for personalized treatment.

## Materials and methods

### Data collection

Transcriptomic expression matrices and corresponding clinical baseline information from both normal control (NC) and STAD tissue samples were obtained from two publicly available databases: The Cancer Genome Atlas (TCGA) and the Gene Expression Omnibus (GEO). Transcriptomic profiles (count format) and clinical data for STAD samples from TCGA were downloaded in the R programming environment and "limma" R script was employed to convert the count-format transcriptomic data obtained from the TCGA database into transcripts per million (TPM) formats. Gene annotation and parsing of transcriptomic matrices were conducted using Perl, based on the human genome annotation files. Samples lacking survival information or with an overall survival (OS) rate of less than 30 days were excluded from further analysis. Following this filtering, a total of 32 NC samples and 337 STAD samples from the TCGA cohort were included in subsequent analyses. From the GEO database, two STAD-related expression datasets were selected and downloaded: GSE84437 and GSE15459. Transcriptomic data from these datasets were annotated and normalized using platform-specific annotation file (GSE84437: GPL6947, Illumina HumanHT-12 V3.0 Expression BeadChip; GSE15459: GPL570, [HG-U133_Plus_2] Affymetrix Human Genome U133 Plus 2.0 Array) in the Perl environment. After similarly excluding samples lacking survival information or with an OS less than 30 days, 431 STAD samples from GSE84437 and 182 STAD samples from GSE15459 were retained. For integrative analysis, the TCGA and GSE84437 datasets were merged using the "limma" package in R, forming the training cohort [30]. Batch effects and data normalization were corrected using the "sva" package [7]. The GSE15459 dataset was designated as an independent external validation cohort. Additionally, data on tumor mutation burden (TMB) and copy number variation (CNV) for STAD samples were obtained from the UCSC Xena database for further analysis.

### Identification of differentially expressed parthanatos-related genes and molecular subtype characterization

A total of 37 PRG were retrieved from the GeneCards database using "Parthanatos" as the search keyword (S1 Table). Differential expression analysis between NC and STAD samples was performed based on the criteria of |fold change| ≥ 2 and an adjusted $p$-value < 0.05 via "limma" R script. Differentially expressed PRG (DE-PRG) were visualized using the "ggplot2" package in R. Additionally, the mutation frequency of DE-PRG in STAD samples was illustrated with a waterfall plot generated using the "maftools" package. To further explore molecular heterogeneity, unsupervised consensus clustering was performed based on the expression profiles of DE-PRG using the "pam" algorithm implemented in the "ConsensusClusterPlus" package. The number of clusters (k) was evaluated across a range of 2 to 9. Consensus heatmaps, cumulative distribution function (CDF) curves, and delta area plots were generated to determine the optimal clustering solution. To assess transcriptional differences among molecular subtypes, principal component analysis (PCA) was conducted using the "ggplot2" package. Kaplan–Meier survival analysis was performed to evaluate differences in overall survival among the identified subtypes, with statistical significance assessed using the log-rank test via the "survival" package. Furthermore, gene set variation analysis (GSVA) was employed to investigate pathway activity differences across PRG subtypes. KEGG pathway gene sets were sourced from the "c2.cp.kegg.v7.2.symbols.gmt" file, and GSVA enrichment scores were calculated using the "GSVA" package in R.

### Immune microenvironment infiltration landscape and immunotherapy response prediction

Based on the transcriptomic matrix of STAD samples, the ESTIMATE algorithm (Estimation of STromal and Immune cells in MAlignant Tumor tissues using Expression data) was applied using the "estimate" R package to infer the infiltration

levels of stromal and immune components within the tumor microenvironment. This analysis yielded four quantitative indices: Immune Score, Stromal Score, ESTIMATE Score, and tumor purity, providing a comprehensive assessment of immune infiltration across molecular subgroups. To further quantify the relative abundance of 23 immune cell types, single-sample Gene Set Enrichment Analysis (ssGSEA) was conducted using the "GSVA" package in R, based on curated marker genes specific to each immune cell population. To evaluate the potential responsiveness of different subgroups to immunotherapy, the Tumor Immune Dysfunction and Exclusion (TIDE) algorithm was employed via the TIDE web platform (http://tide.dfci.harvard.edu/). The TIDE score integrates features of T cell dysfunction and T cell exclusion to simulate mechanisms of tumor immune evasion, thereby predicting potential responses to immune checkpoint inhibitors targeting PD-1/PD-L1 and CTLA-4 pathways. In addition, immune-related scores from The Cancer Immunome Atlas (TCIA; https://tcia.at/) were used to further assess immunotherapeutic potential. Specifically, the Immunophenoscore (IPS) was compared among subgroups to evaluate differential sensitivities to PD-1 and CTLA-4 blockade.

### Identification of gene subtype characterization associated with prg molecular subtypes

Differentially expressed genes (DEGs) among the PRG molecular subtypes were identified using the "limma" package in R, with threshold criteria set at |fold change| ≥ 2 and adjusted $p$-value < 0.05. To investigate potential molecular regulatory mechanisms underlying these differences, Gene Ontology (GO) and Kyoto Encyclopedia of Genes and Genomes (KEGG) enrichment analyses were performed using the "clusterProfiler" R package. Based on the expression profiles of the identified DEGs, unsupervised consensus clustering was conducted with the "pam" algorithm implemented in the "ConsensusClusterPlus" package to classify STAD samples into distinct gene subtypes. PCA was then performed using the "ggplot2" package to visualize gene subtype distribution and separation. Kaplan-Meier survival analysis was carried out using the log-rank test via the "survival" package to evaluate the prognostic significance of these subtypes. Furthermore, the expression patterns of DEGs across PRG molecular subtypes, gene subtypes, and different clinicopathological parameters were visualized using the "pheatmap" package in R.

### Construction of prognostic risk model based on the prg score using integrated machine learning algorithms

Univariate Cox regression analysis was performed using the "survival" R package to assess the prognostic significance of DEGs within the training datasets (TCGA and GSE84437). Using the GSE15459 dataset as an independent validation cohort, we employed a leave-one-out cross-validation (LOOCV) framework to implement 100 algorithm combinations derived from 10 distinct machine learning algorithms (CoxBoost, Enet, GBM, LASSO, plsRcox, Ridge, RSF, stepwise Cox, SuperPC and survival-SVM) [31]. The concordance index (C-index) was calculated to evaluate the predictive performance of each model. Prognostic feature genes were selected based on the model yielding the highest C-index. These features were then subjected to multivariate Cox regression analysis to establish a prognostic risk model based on the PRG score. Patients in both the training and validation cohorts were stratified into high- and low-risk subgroups according to the median PRG score. To illustrate the relationships among molecular subtypes, PRG score groups, and clinical outcomes, a Sankey diagram was generated using the "ggalluvial" R package.

### Independent prognostic analysis and nomogram construction

Univariate and multivariate Cox regression analyses were performed using the "survival" R package to estimate the hazard ratios (HR) and P-values for each clinical variable and the PRG score in both the training and validation cohorts. To predict 1-, 3-, and 5-year OS rate in patients with STAD, a prognostic nomogram integrating clinical variables and the PRG score was constructed using the "rms" R package. Calibration curves were generated with the "regplot" and "rms" R packages to assess the consistency between the predicted and observed survival probabilities. Time-dependent receiver operating characteristic (ROC) curves were plotted using the "timeROC" R package, and the area under the curve (AUC) was calculated to evaluate the predictive accuracy of the model.

## Tumor mutation burden landscape and drug sensitivity prediction

TMB data for STAD samples were extracted using Perl scripts, and TMB scores were calculated for each individual sample. The somatic mutation profiles of different PRG score subgroups were visualized using waterfall plots generated by the "maftools" R package. Furthermore, to assess the potential sensitivity to chemotherapeutic agents, the half-maximal inhibitory concentration (IC50) values were predicted across subgroups using the "pRRophetic" R package, based on data from the Genomics of Drug Sensitivity in Cancer (GDSC) database.

## Single-Cell RNA sequencing analysis

The single-cell RNA sequencing dataset GSE163558 (10x) from the GEO database, which includes three STAD and one NC sample, was analyzed. Raw expression matrices were processed in the R environment (version 4.4.1) using the Seurat package (version 4.3.0). Cells with fewer than 500 or more than 5,000 unique molecular identifiers (UMIs), or with mitochondrial gene percentages exceeding 10%, were excluded to filter out low-quality cells. Following quality control, the data were normalized using the "LogNormalize" method, and the top 2,000 highly variable genes were identified using the "FindVariableFeatures" function. Gene expression data were scaled with the "ScaleData" function, followed by dimensionality reduction via PCA plot. The top 20 principal components, selected based on the ElbowPlot and JackStraw test results, were retained for downstream analysis. To correct for batch effects, the "Harmony" function was applied, and data integration was carried out using the "IntegrateData" function. A cell-neighbor graph was constructed using the "FindNeighbors" function, and cell clustering was performed with the Louvain algorithm at a resolution of 1.2 via the "FindClusters" function. Differentially expressed genes among clusters were identified using the "FindMarkers" function with thresholds set at min.pct = 0.25 and logfc.threshold = 0.25 (adjusted $p < 0.05$). Dimensionality reduction was visualized using UMAP and t-SNE. Cell types were annotated using the "SingleR" algorithm with the HumanPrimaryCellAtlasData reference and manually validated using the CellMarker database. All visualizations, including UMAP plots, heatmaps, and violin plots, were generated using Seurat and ggplot2.

## Cell-cell communication analysis

To explore intercellular communication patterns between different cell types, we applied the "CellChat" R package to model ligand-receptor interactions using the processed Seurat object. The dataset was first split into GC and NC groups using the "SplitObject" function. CellChat objects were constructed for each group using createCellChat, and cell populations were annotated using the "cellType" metadata. We then specified the human ligand-receptor interaction database CellChatDB.human and performed a series of preprocessing steps, including subsetData, identifyOverExpressedGenes, and identifyOverExpressedInteractions, to extract relevant signaling molecules. Merged CellChat objects were used to compare communication patterns across groups with compareInteractions, netVisual_circle, and netVisual_diffInteraction. Interaction heatmaps (netVisual_heatmap) were plotted to visualize frequency distributions. To investigate functional differences in signaling, we applied computeNetSimilarityPairwise, followed by netEmbedding and rankSimilarity. All visualizations were generated using ggplot2-based plotting functions and circle or heatmap layouts provided by CellChat.

## Cell culture

The human gastric epithelial cell line GES-1 and the human gastric cancer cell line HGC-27 were purchased from the American Type Culture Collection (ATCC). GES-1 cells were cultured in RPMI-1640 medium supplemented with 10% fetal bovine serum (FBS), while HGC-27 cells were maintained in DMEM containing 10% FBS. Both cell lines were incubated at 37°C in a humidified atmosphere with 5% $CO_2$. Cells were passaged when they reached 70–80% confluency using 0.25% trypsin. The culture medium was replaced every 2–3 days. Cell morphology and growth were regularly monitored under an inverted microscope to ensure optimal conditions and to check for any signs of contamination.

## Western blot analysis

Total protein was extracted from GES-1 and HGC-27 cells using RIPA lysis buffer supplemented with 1% PMSF. Cells were cultured to 80–90% confluence, washed with PBS, and lysed on ice for 15 minutes. The lysates were sonicated and then centrifuged at $12,000 \times g$ for 10 minutes at 4°C. Protein concentrations were determined using a BCA assay (Beyotime, P0012). Equal amounts of protein (20 μg) were separated on 12% SDS-PAGE gels and transferred onto PVDF membranes (Millipore, IPVH00010). The membranes were blocked with 5% non-fat milk for 1 hour and incubated overnight at 4°C with primary antibodies against COL8A1 (1:1000, Abcam, ab192350) and GAPDH (1:5000, Sigma, A1978). After washing with TBST, the membranes were incubated with HRP-conjugated secondary antibody (1:5000, CST, 7074) for 1 hour. Chemiluminescent detection was performed using ECL reagent (Thermo Fisher, 34095), and images were captured with the Chemi-Doc XRS+ system (Bio-Rad). Band intensities were quantified using ImageJ software, with GAPDH as the loading control.

## siRNA-mediated knockdown of COL8A1

To investigate the role of COL8A1 in HGC-27 cells, small interfering RNA (siRNA) was used for gene silencing. COL8A1-specific siRNA sequences were designed and synthesized by GenePharma. A non-targeting siRNA (siNC) was used as the negative control. HGC-27 cells were seeded in 6-well plates to achieve 70–80% confluence. Transfection was carried out using Lipofectamine™ 3000 (Thermo Fisher, L3000-015) according to the manufacturer's protocol. siRNA and Lipofectamine were pre-incubated at room temperature for 10–15 minutes to form complexes, which were then added to the cells. After 24 hours, the medium was replaced, and the cells were cultured for an additional 48 hours. The efficiency of COL8A1 knockdown was confirmed by Western blot analysis of COL8A1 protein expression.

## Cell viability assay

To evaluate the proliferation of HGC-27 cells transfected with either siNC or siCOL8A1, a CCK-8 assay was performed using the CCK-8 kit (Dojindo, CK04). Transfected cells were seeded into 96-well plates at a density of 1,000 cells per well. After 24 hours, 10 μL of CCK-8 solution was added to each well and incubated for 2 hours. The optical density was measured at 450 nm using a microplate reader (BioTek, ELx800). Measurements were taken at 24-, 48-, 72-, and 96-hours post-transfection. Each group was tested in triplicate.

## Colony formation assay

To assess the impact of COL8A1 knockdown on clonogenic capacity, transfected HGC-27 cells were seeded at a density of 200 cells per well in 6-well plates and cultured for 1 week. The medium was refreshed every 3 days. Colonies were fixed in 4% paraformaldehyde and stained with 0.1% crystal violet for 1 hour. After washing, colonies containing ≥50 cells were counted under a microscope. All assays were performed in triplicate.

## Transwell migration and invasion assays

To evaluate the effects of siNC and siCOL8A1 transfection on the migration and invasion capabilities of HGC-27 cells, Transwell chambers (Corning, 3422) were used. For the migration assay, transfected HGC-27 cells ($2 \times 10^4$ cells/well) were suspended in serum-free RPMI-1640 medium and seeded into the upper chambers. The lower chambers were filled with medium containing 20% fetal bovine serum as a chemoattractant. After 24 hours of incubation, non-migrated cells on the upper surface of the membrane were gently removed with a cotton swab. Migrated cells on the underside of the membrane were fixed with 4% paraformaldehyde, stained with 0.1% crystal violet, and then counted under a microscope. For the invasion assay, Transwell chambers were pre-coated with Matrigel (BD Biosciences). The rest of the procedure was the same as for the migration assay. The number of invading cells was quantified by counting stained cells in five randomly selected fields under a microscope. Each experiment was performed in triplicate.

 

## Statistical analysis

In this study, all statistical analyses were conducted using R software (version 4.4.1), Perl, and GraphPad Prism. Kaplan-Meier survival curves were generated to assess overall survival, and differences between groups were compared using the log-rank test. The Wilcoxon rank-sum test and Student's t-test were used to evaluate statistical significance between two groups, while one-way analysis of variance (ANOVA) was employed for comparisons among multiple groups. For multiple comparisons, the Bonferroni correction was applied to adjust $p$-values. All statistical tests were two-sided, and a $p$-value $< 0.05$ was considered statistically significant. Data are presented as mean $\pm$ standard deviation (SD). Statistical significance is indicated as follows: * $p < 0.05$, ** $p < 0.01$, *** $p < 0.001$.

## Results

### Differential expression analysis and mutation landscape of PRG signatures in STAD

A total of 37 PRGs were analyzed to investigate their potential regulatory roles in the initiation and progression of STAD. The differential expression analysis revealed that 14 PRG signatures were significantly upregulated in the STAD group compared to the NC group, such as COL8A1, FEN1, NAT10, CYBB, and PARP1 (Fig 1A). CNV analysis revealed that most PRG genes exhibited significant copy number amplifications in STAD, particularly TOMM20, DDB1, COL8A1, and FEN1, whereas DIABLO and ESR2 showed notable deletions (Fig 1B). Tumor mutation burden analysis indicated that in 433 STAD samples, 73 samples showed altered mutation frequencies in PRG signatures, with mutation rates of 4% for COL8A1, 3% for DDB1, 3% for PARP1, and 3% for ESR2 (Fig 1C). Furthermore, we assessed the association of these DE-PRG signatures with STAD prognosis. Network analysis revealed that most DE-PRG signatures were significantly positively correlated, and COL8A1 and SQSTM1, as risk factors, were associated with poor prognosis in STAD (Fig 1D). Based on these results, we found that the expression of PRG signatures in STAD was significantly different and associated with genetic variations and prognosis.

### Molecular subtype characterization of PRG in STAD

To further clarify the potential role of PRG in STAD, we integrated the TCGA and GSE84437 datasets as the training set and extracted 768 STAD samples to identify PRG molecular subtypes. Based on the optimal parameters of the unsupervised consensus clustering analysis model (k = 3), we accurately identified three distinct PRG molecular subtypes in STAD samples. PRG subtype A included 328 samples, subtype B included 256 samples, and subtype C included 184 samples (Fig 2A-2C). PCA revealed significantly distinct distribution patterns between the three PRG molecular subtypes (Fig 2D). Using log-rank analysis, we evaluated the clinical survival outcomes of the three PRG molecular subtypes, and the results indicated that PRG subtype C had the poorest clinical prognosis (Fig 2E). Differential expression analysis revealed significant expression differences in PRG signatures among the three subtypes, particularly in the worst prognosis subtype C, where genes like COL8A1 and CYBB were significantly overexpressed (Fig 2F). GSVA assessed the differential regulation of KEGG signaling pathways across the PRG molecular subgroups. Compared to PRG subtype A, we found that immune-related pathways, such as the chemokine signaling pathway, cytokine-cytokine receptor interaction, leukocyte transendothelial migration, and cell-adhesion-molecules, were significantly downregulated in PRG subtype B. In contrast, in the poorest prognosis PRG subtype C, we observed significant upregulation of tumor-related signaling pathways, such as pathways in cancer and TGF-beta signaling pathway. Notably, immune-related pathways, including cell-adhesion-molecules, leukocyte transendothelial migration, and ECM-receptor interaction, were significantly upregulated in PRG subtype C (Fig 2G, 2H). Compared to PRG subtype A, mismatch repair and DNA replication signaling pathways were significantly downregulated in PRG subtype C, while tumor-related pathways such as WNT signaling pathway, pathways in cancer, and basal cell carcinoma were significantly upregulated (Fig 2I). Based on these results, we found that STAD samples could be accurately classified into three distinct molecular subtypes based on PRG signatures, which

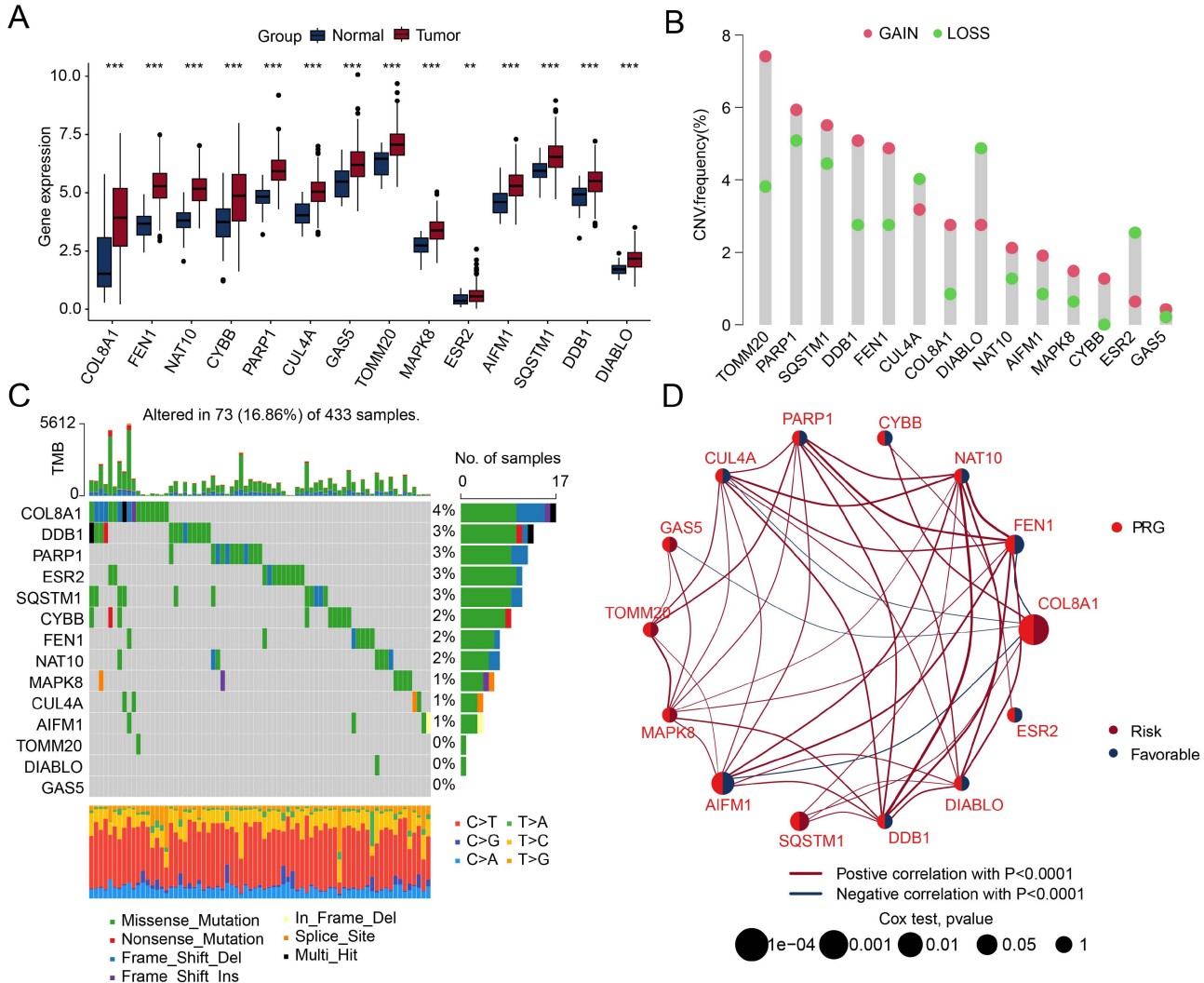

**Fig 1. Differential expression analysis and mutational landscape of PRG.** (A) Differential expression analysis of PRG between NC and STAD samples. Thresholds for significance were set at |fold change| ≥ 2 and adjusted p-value < 0.05. (B) CNV frequency analysis of DE-PRG signatures in STAD. (C) Waterfall plot illustrating the mutation burden landscape of DE-PRG signatures in STAD. (D) Prognostic significance and correlation analysis of DE-PRG signatures in STAD.

were associated with clinical prognosis. Additionally, the abnormal regulation of immune-related pathways may be a key regulatory factor underlying the clinical prognosis differences in PRG molecular subtypes.

## Immune microenvironment infiltration landscape and immunotherapy response prediction of PRG molecular subtypes

GSVA results revealed aberrant regulation of immune-related pathways across the three PRG molecular subtypes, suggesting that immune dysregulation may contribute to the observed differences in clinical outcomes. Subsequently, using various immune infiltration algorithms, we assessed the immune microenvironment infiltration landscape of the PRG molecular subtypes. ESTIMATE results indicated that in PRG subtype B, the immune score, stromal score, and

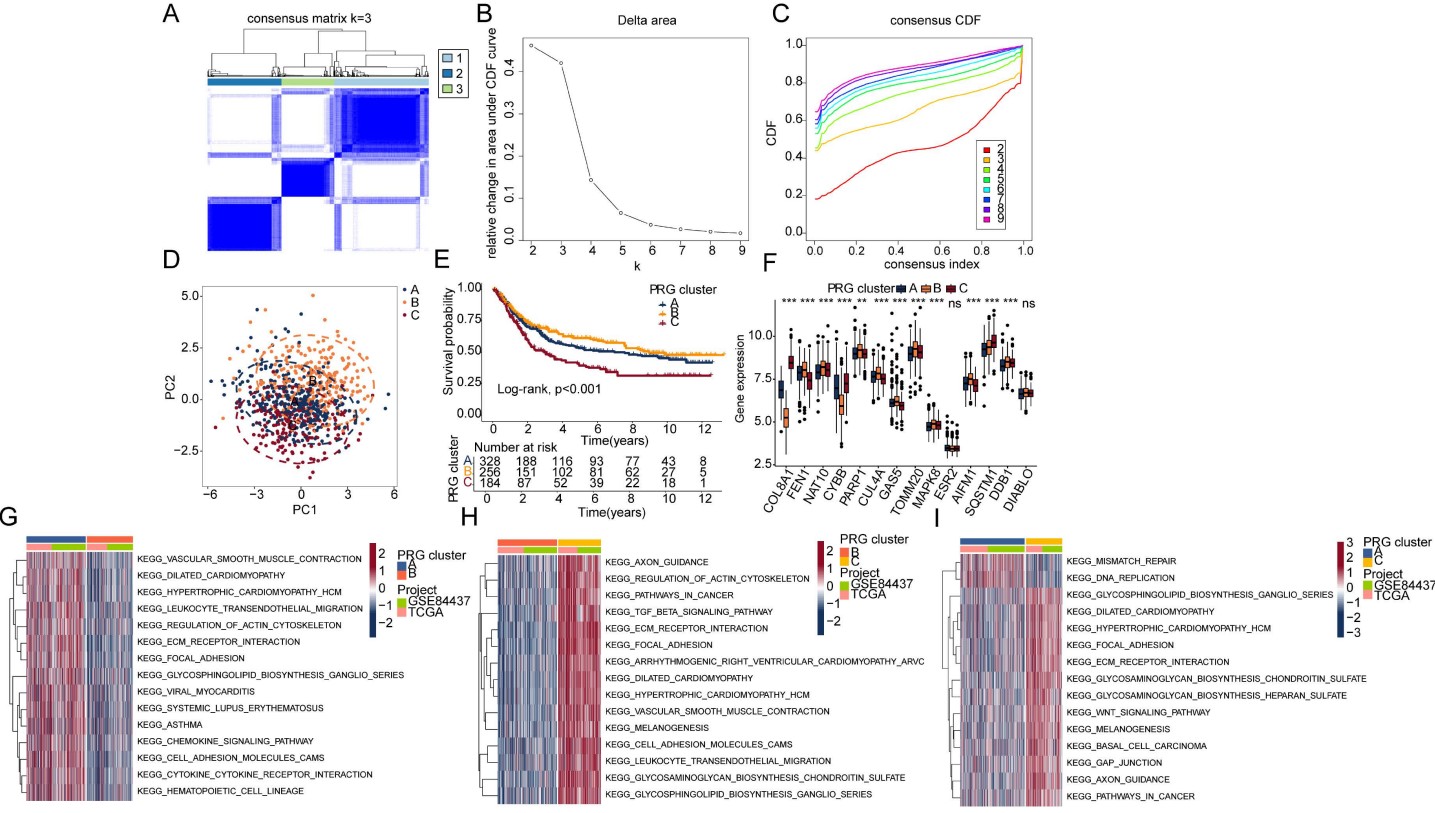

**Fig 2. Identification of PRG molecular subtypes and prognostic outcome analysis.** (A-C) Identification of distinct PRG molecular subtypes in STAD using unsupervised consensus clustering analysis. (D) PCA plot illustrating the distinct distribution patterns of the identified PRG subtypes. (E) Kaplan–Meier survival analysis based on the log-rank test to evaluate the prognostic outcomes of different PRG subtypes. (F) Differential expression analysis of DE-PRG signatures among the PRG molecular subtypes. (G-I) KEGG pathway enrichment analysis showing signaling differences among PRG molecular subtypes.

ESTIMATE score of STAD samples were significantly reduced, while tumor purity was markedly increased, suggesting that PRG subtype B might exhibit a prominent immunosuppressive state (Fig 3A-3D). Using ssGSEA analysis, we quantitatively evaluated the infiltration proportions of 23 immune cell types in the three PRG subtypes. The results showed that in PRG subtype B, the infiltration of immune cells, such as activated B cells, activated CD8 T cells, activated dendritic cells, immature B cells, immature dendritic cells, and MDSCs, was significantly reduced (Fig 3E). TIDE scores indicated that in PRG subtype C, the TIDE score was significantly higher than in subtypes A and B, suggesting poorer efficacy of immune checkpoint inhibitors in this subtype (Fig 3F). Moreover, IPS scores indicated that PRG subtype C had significantly lower IPS scores, implying a poorer response to CTLA4 and PD1 immune therapies (Fig 3G-3I). Based on these results, we infer that there are significant differences in the immune infiltration states among the three PRG molecular subtypes, and a higher immune infiltration state might be associated with poor prognosis in STAD samples.

## Identification of gene subtype related to PRG molecular subtypes

To elucidate the potential mechanisms behind the clinical survival outcome differences of PRG molecular subtypes, we identified DEGs between the PRG subgroups under the threshold conditions of |fold change| ≥ 2 and p.adjust < 0.05 (Fig 4A). GO enrichment analysis revealed that these 71 DEGs were associated with biological functions such as regulation

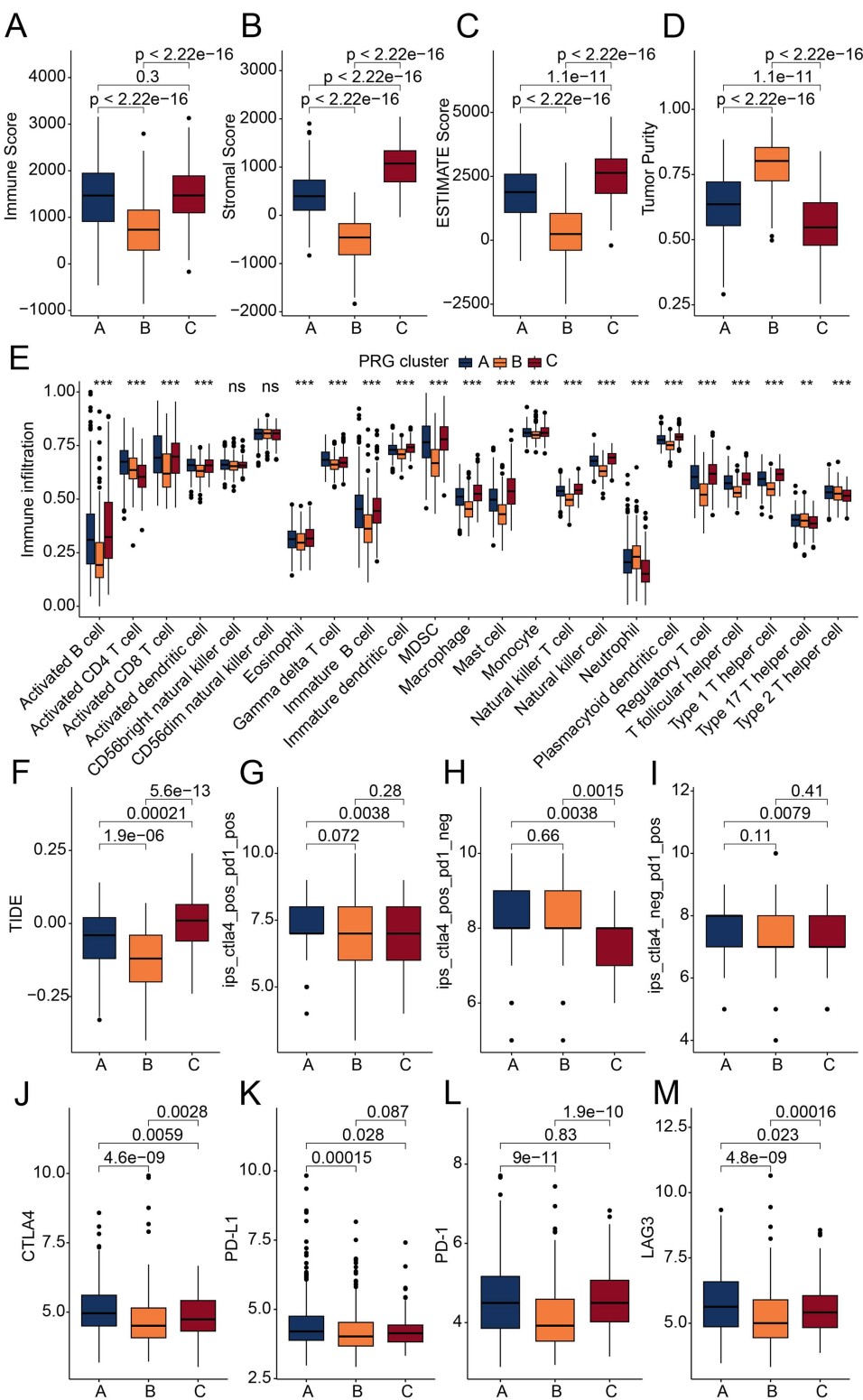

**Fig 3. Immune microenvironment landscape and immunotherapy response evaluation of PRG molecular subtypes.** (A–D) Immune infiltration status of PRG molecular subtypes assessed using the ESTIMATE algorithm. (E) Quantitative analysis of the infiltration levels of 23 immune cell types based on the ssGSEA algorithm. (F) TIDE score-based prediction of immune evasion potential. (G–I) IPS score analysis revealing the differential response of PRG molecular subtypes to CTLA-4 and PD-1 targeted therapies.

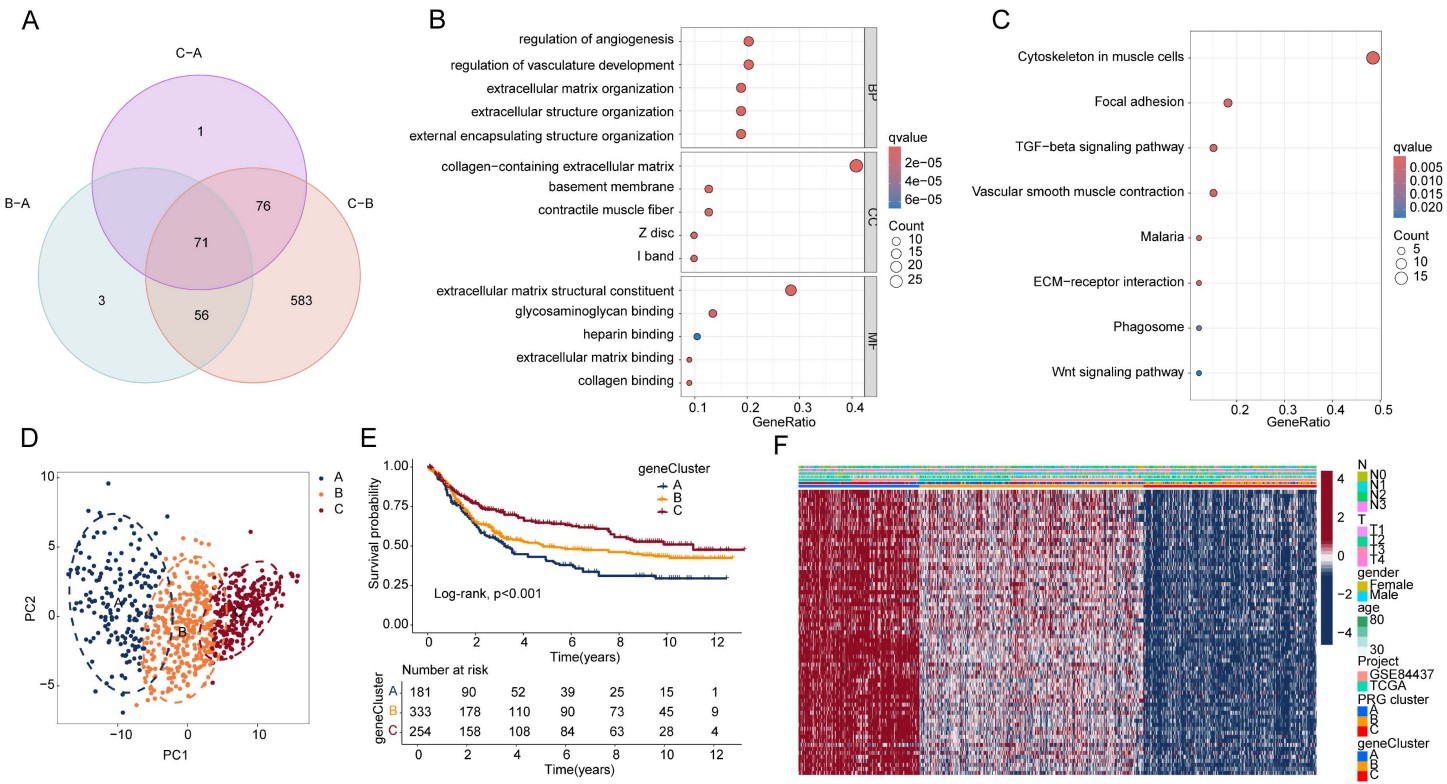

**Fig 4. Gene subtyping analysis based on DEGs among PRG molecular subtypes.** (A) Identification of DEGs among the PRG molecular subtypes. (B, C) GO and KEGG enrichment analyses of the identified DEGs. (D) PCA plot illustrating the distribution patterns of the three gene subtypes. (E) Kaplan–Meier survival curves of the gene subtypes based on log-rank analysis. (F) Expression profiles of DEGs across different clinicopathological features and molecular subgroups.

of vasculature development, regulation of angiogenesis, collagen-containing extracellular matrix, and extracellular matrix structural constituent. KEGG enrichment analysis indicated that the DEGs were involved in signaling pathways regulating cytoskeleton in muscle cells, focal adhesion, and TGF-beta signaling pathway (Fig 4B, 4C). Based on the expression characteristics of these DEGs, we applied an unsupervised consensus clustering algorithm for secondary molecular subtype analysis of STAD samples. Using the optimal parameters, we classified the STAD samples into three gene subtypes. PCA analysis revealed significantly separated distribution patterns among the three gene subtypes, highlighting the independence of each subtype (Fig 4D). Clinical survival curve analysis demonstrated significant differences in survival outcomes between the three gene subtypes, with gene subtype C exhibiting significantly better survival compared to subtypes B and A (Fig 4E). Expression profile analysis revealed the expression patterns of DEGs in different clinical pathological features and molecular subgroups. The results suggested that in gene subtype C, which had the best prognosis, the expression of DEGs was significantly lower (Fig 4F).

## Comprehensive analysis of PRG score index for predicting clinical prognosis of STAD based on machine learning algorithms

In the training cohort, we used univariate Cox analysis to assess the prognostic value of DEGs in STAD and identify prognostic variables. Based on an independent external validation cohort (GSE15459 dataset), we developed 100 algorithmic combinations using 10 machine learning methods in a LOOCV framework and calculated the C-index for each machine

learning algorithm combination in the training set. The C-index results indicated that the StepCox[both]+Enet[alpha = 0.1] algorithm combination was optimal, and we used multivariate Cox analysis to calculate the PRG score index in both the training and validation sets. Using the median PRG score, we stratified STAD patients into high- and low PRG score subgroups in both the training and validation cohorts (Fig 5B, 5C). Clinical survival curve analysis indicated that in both the training and validation sets, the low PRG score index subgroup had significantly better clinical survival outcomes compared to the high PRG score index subgroup (Fig 5D, 5E). Among the PRG molecular and gene subtypes, we found that in the subgroup with the best prognosis, the PRG score was significantly higher, suggesting that a higher PRG score is associated with poor prognosis in STAD (Fig 5F, 5G). Furthermore, Sankey diagram results showed that in both the PRG molecular and gene subgroups, the better prognosis subgroups tended to have lower PRG scores, which were associated with better clinical survival outcomes (Fig 5H).

## Independent prognostic value assessment and nomogram model construction

In subsequent studies, we further evaluated the independent prognostic value of the PRG score index in predicting STAD clinical survival outcomes. In the training cohort, univariate and multivariate results indicated that stage and PRG score were associated with poor prognosis in STAD (Fig 6A, 6B). In the validation cohort (GSE15459), univariate and multivariate Cox analysis results showed that age, N stage, and PRG score were associated with poor prognosis in STAD (Fig 6C, 6D). Additionally, based on clinical pathological variables and the PRG score index, we constructed independent nomogram models in both the training and validation sets to assess the 1-year, 3-year, and 5-year survival probabilities of STAD samples (Fig 6E, 6H). Calibration curve results indicated that in both the training and validation sets, the nomogram model predicted survival probabilities for 1-year, 3-year, and 5-year survival, showing good consistency with the actual survival probabilities (Fig 6F, 6I). Time-dependent ROC curve results showed AUC values of 0.704, 0.670, and 0.707 for 1-year, 3-year, and 5-year survival in the training set, and 0.646, 0.680, and 0.683 in the validation set (Fig 6G, 6J). Based on these results, we found that compared to other clinical pathological features, the PRG score index can serve as an independent prognostic factor for reflecting the clinical prognosis of STAD. Moreover, the nomogram model based on clinical pathological features and the PRG score index can accurately assess the survival probabilities of STAD samples over different years.

## Mutation burden landscape and immune therapy response prediction of PRG score subgroups

TMB has been reported to be closely associated with the efficacy of immune checkpoint inhibitors and can serve as a biomarker to predict the response to immunotherapy. In subsequent studies, we further assessed the mutation burden landscape and immune therapy response outcomes in different PRG score subgroups. The mutation burden landscape results showed that in the low PRG score subgroup, TMB scores were significantly higher (Fig 7A). Immune therapy response analysis indicated that compared to the high PRG score subgroup, the IPS score was significantly higher in the low PRG score subgroup, suggesting that the low PRG score subgroup may have better clinical therapeutic benefits when receiving CTLA4 and PD1 immunotherapy (Fig 7B-7D). Mutation burden results showed that 89.6% of samples in the low PRG score subgroup had somatic mutations, while 83.97% of samples in the high PRG score subgroup had significant mutations (Fig 7E, 7F). Notably, in the low PRG score subgroup, the mutation rates of genes such as MUC16 (35%), LRP1B (28%), SYNE1 (25%), and FLG (22%) were significantly elevated.

## Immune microenvironment infiltration landscape and drug sensitivity analysis of PRG score subgroups

ESTIMATE results indicated that in the high PRG score subgroup, the ESTIMATE score and stromal score were significantly higher, while tumor purity was significantly lower. Notably, there were no significant differences in immune scores between PRG subgroups (Fig 8A-8D). ssGSEA analysis results showed that in the low PRG score subgroup,

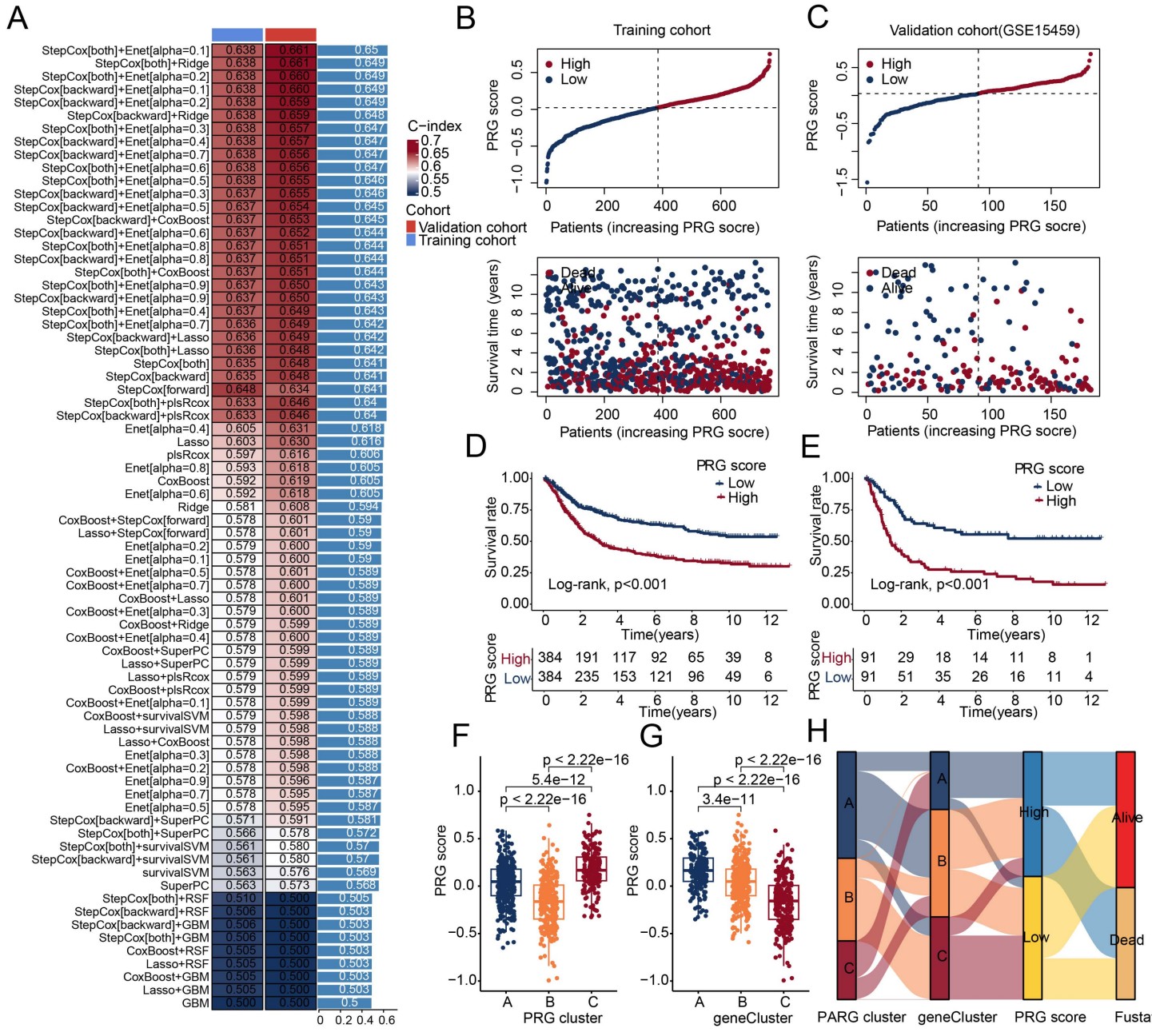

**Fig 5. Construction of the PRG score index using integrated machine learning algorithms.** (A) C-index calculated for both training and validation cohorts using combinations of 10 different machine learning algorithms. (B, C) Stratification of PRG score subgroups in the training and validation cohorts. (D, E) Kaplan–Meier survival analysis of PRG score subgroups in two independent cohorts. (F, G) Differential analysis of PRG scores across PRG score subgroups and gene subtypes. (H) Sankey diagram illustrating the potential relationships among PRG molecular subtypes, gene subtypes, PRG score subgroups, and clinical outcomes.

the infiltration proportion of activated CD4 T cells and neutrophils was significantly higher, while the infiltration proportion of Immature dendritic cells, Macrophages, Mast cells, Natural killer T cells, Natural killer cells, and Plasmacytoid dendritic cells were significantly lower (Fig 8E). Based on the GDSC database, we predicted potential antitumor drugs

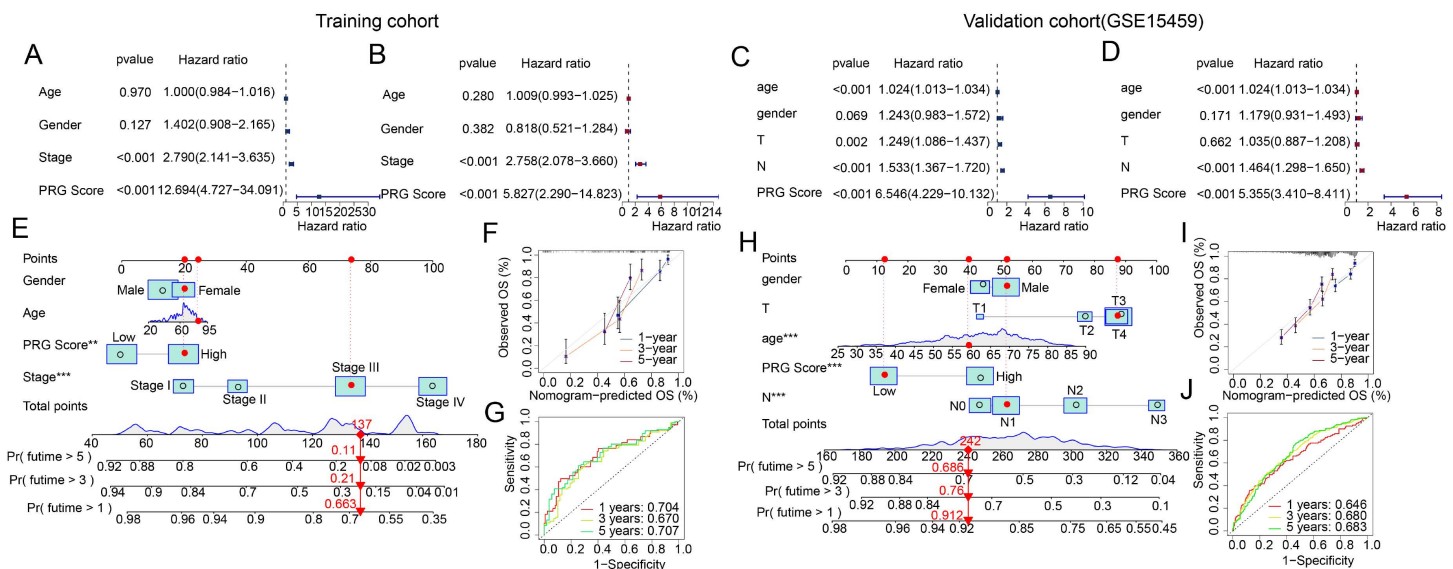

**Fig 6. Evaluation of the independent prognostic value of the PRG score index and construction of a nomogram model.** (A-D) Univariate and multivariate Cox regression analyses of clinicopathological variables and the PRG score in the training cohort and validation cohort. (E, F) Nomogram model construction and calibration curve analysis based on clinicopathological variables and the PRG score index in the training cohort. (G) Time-dependent ROC curve analysis in the training cohort. (H, I) Nomogram model construction and calibration curve analysis based on clinicopathological variables and the PRG score index in the validation cohort. (J) Time-dependent ROC curve analysis in the validation cohort.

with therapeutic response in PRG score subgroups. IC50 results indicated that in the high PRG score subgroup, the IC50 values for Saracatinib and Pazopanib were significantly lower, suggesting that the high PRG score subgroup might have better drug treatment response. Additionally, the low PRG score subgroup may have a better drug treatment response when receiving Ruxolitinib, Roscovitine, Rapamycin, 5-Fluorouracil, Phenformin, and Ispinesib Mesylate (Fig 8F).

## Single-cell sequencing analysis reveals expression features of prognostic signatures in cell subpopulations

We further assessed the classification of cell subpopulations and the expression characteristics of prognostic signatures in STAD at the single-cell sequencing level. Based on the single-cell dataset GSE163558 of STAD samples, we extracted single-cell sequencing data from 1 normal sample and 3 STAD samples for analysis. After quality control and normalization of the scRNA data from the 4 samples, we identified 2000 highly variable genes for dimensionality reduction analysis (Fig 9A, 9B). The "Harmony" algorithm was used to standardize the 4 samples and eliminate batch effects between them (Fig 9C). Using marker genes, we identified 22 cell types and employed UMAP and tSNE dimensionality reduction analysis to reveal the distribution characteristics of each cell type (Fig 9D, 9E). Violin plots indicated that the prognostic signatures such as COL8A1, PLXDC2, and CCDC80 were highly expressed in 22 cell types (Fig 9F). Using the SingleR annotation algorithm, we accurately identified 9 cell subpopulations from the 22 cell types: T cells, Neutrophils, B cells, Epithelial cells, Monocytes, Tissue stem cells, NK cells, Endothelial cells, and Macrophages (Fig 9G, 9H). Additionally, we further evaluated the expression of the PRG signature in these 9 cell subpopulations. Violin plot results showed significant expression of the PRG signature in T cells and Neutrophils (Fig 9I). Among the 9 cell subpopulations, we observed that COL8A1 was significantly expressed in Tissue stem cells, while PLXDC2 showed higher expression in Macrophages, Endothelial cells, Tissue stem cells, and Monocytes (Fig 9J).

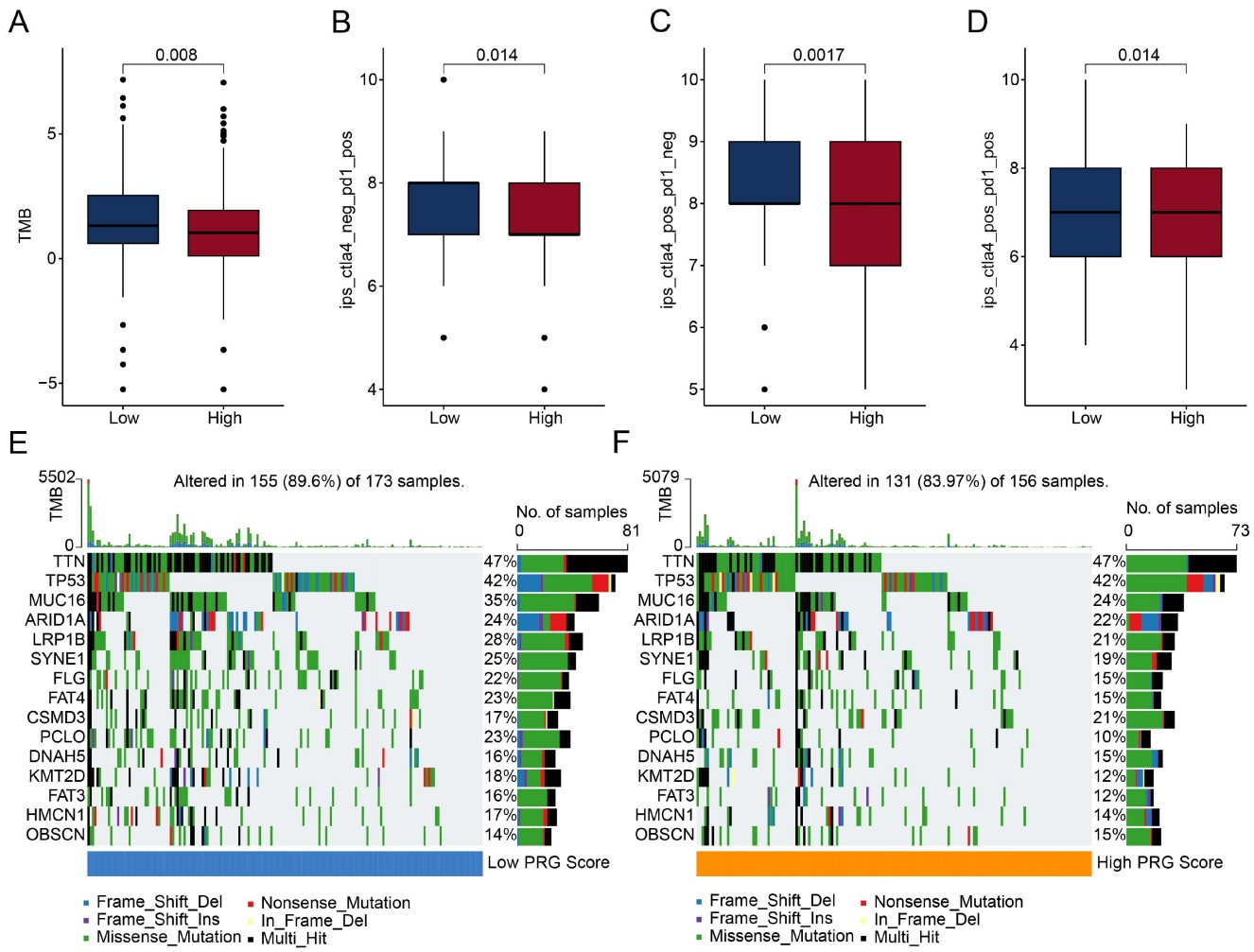

**Fig 7. Immunotherapy response prediction and mutational burden landscape analysis of PRG score subgroups.** (A) Differential analysis of TMB scores between PRG scores subgroups. (B–D) IPS analysis of PRG score subgroups. (E, F) Waterfall plots illustrating the somatic mutation landscape of PRG score subgroups.

## Cell-cell communication landscape

To explore intercellular communication patterns, we constructed comprehensive cell–cell interaction networks based on single-cell transcriptomic profiles. The analysis revealed a marked increase in both the number and overall strength of cell–cell interactions in the GC group compared to the NC group, suggesting enhanced signaling activity within the tumor microenvironment (Fig 10A). Differential interaction network analysis further illustrated that GC exhibited substantial remodeling of intercellular signaling, with particularly enhanced communication observed among tissue stem cells, endothelial cells, and epithelial cells, indicating their potentially central roles in GC progression (Fig 10B). Circle plots visualizing the cellular interaction networks showed denser connections involving monocytes, endothelial cells, and epithelial cells in the GC group, implying that these cell types might possess more active signaling functions under pathological conditions (Fig 10C). Finally, heatmaps quantitatively illustrated the number of inferred interactions between each pair of cell types. Increased communication between epithelial cells and both tissue stem cells and endothelial cells was particularly prominent in GC, suggesting enhanced immune–stromal interactions in the disease state (Fig 10D).

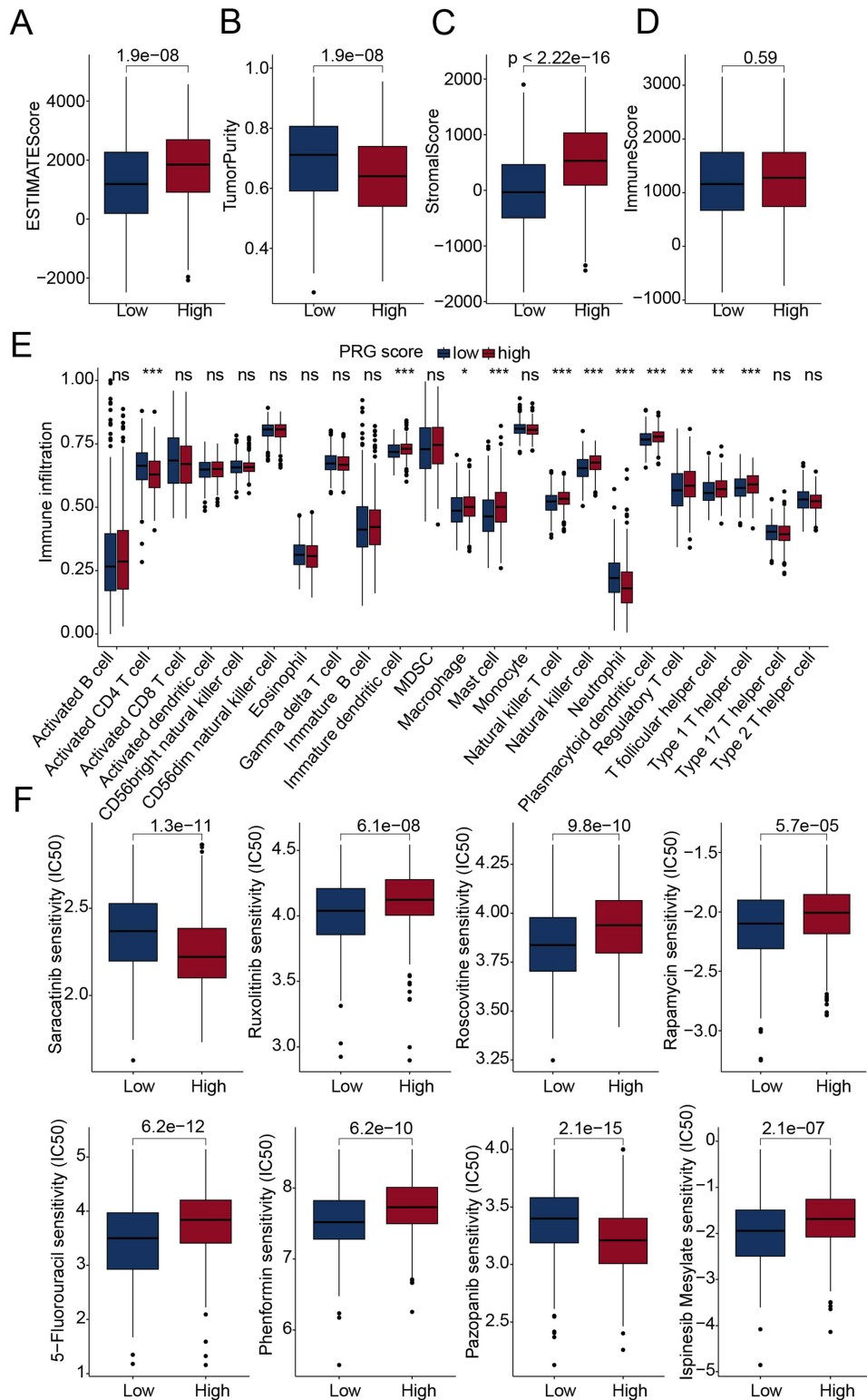

**Fig 8. Immune microenvironment landscape and drug sensitivity analysis.** (A–D) Evaluation of immune infiltration characteristics using the ESTI-MATE algorithm. (E) Assessment of the infiltration proportions of 23 immune cell types based on the ssGSEA algorithm. (F) Drug sensitivity analysis of PRG score subgroups.

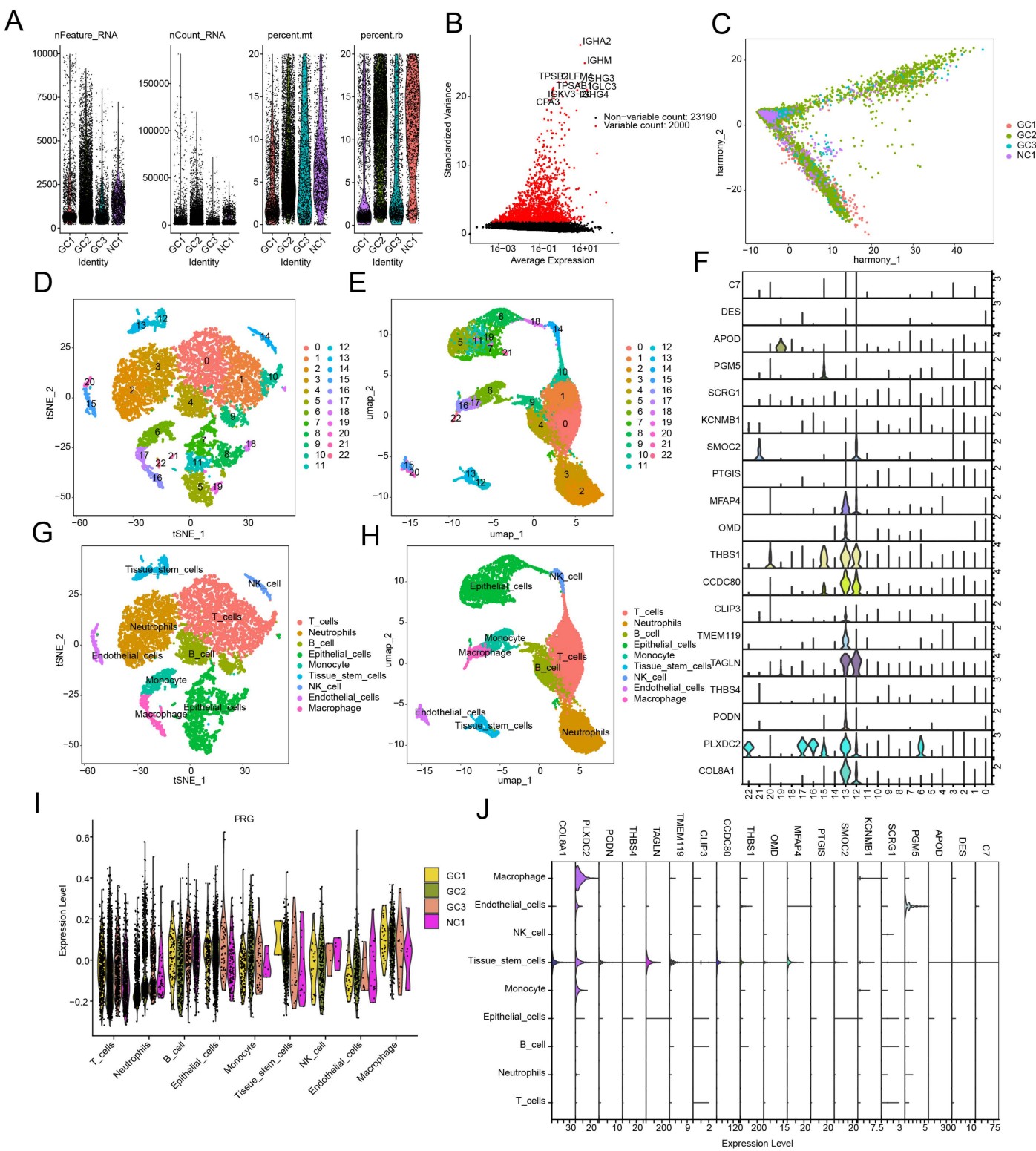

**Fig 9. Single-cell RNA sequencing analysis reveals the expression patterns of the prognostic signature across cellular subpopulations.**
(A) Quality control of the single-cell RNA-seq dataset GSE163558. (B) Identification of the top 2,000 highly variable genes. (C) Batch effect correction and data normalization using the Harmony algorithm. (D, E) UMAP and t-SNE visualizations showing the distribution of 22 distinct cell subtypes. (F)

Expression analysis of the prognostic signature across the 22 identified cell subtypes. (G, H) UMAP and t-SNE plots illustrating the distribution of 9 annotated cell clusters. (I) Quantitative analysis of the PRG signature expression across different cell clusters. (J) Expression profiling of the prognostic signature within the 9 annotated cell subpopulations.

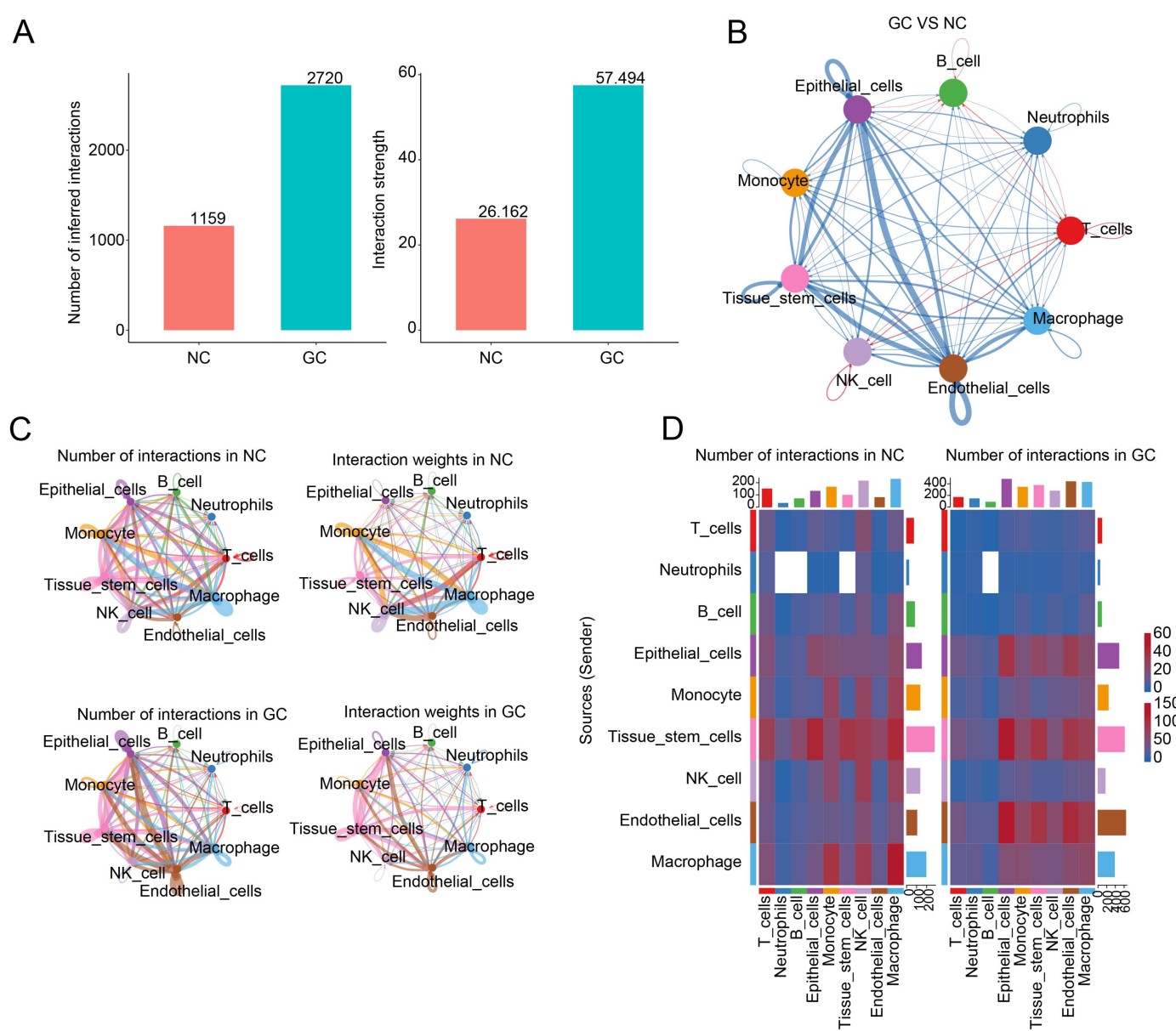

**Fig 10. Comparative analysis of cell-cell communication networks between NC and GC samples.** (A) Bar plots showing the total number (left) and overall strength (right) of inferred intercellular interactions in NC and GC samples. (B) Differential intercellular communication network between GC and NC. (C) Circle plots depicting the number and strength of intercellular communications in NC (top row) and GC (bottom row). (D) Heatmaps displaying the number of interactions among different cell types in NC (left) and GC (right).

## COL8A1 knockdown significantly suppresses the proliferation and migration of STAD cells

Based on the risk coefficients constructed using the PRG scoring system, we found that COL8A1 had the highest risk coefficient, suggesting that it may be a key factor influencing STAD prognosis. Therefore, we further explored the potential role of COL8A1 in the development of STAD. Western blot analysis revealed that COL8A1 protein expression was significantly higher in the HGC-27 cell line compared to the normal GSE-1 cell line (Fig 11A, 11C). To further validate the biological function of COL8A1 in STAD, we used siRNA interference to construct a COL8A1 knockdown cell model. WB analysis showed that siRNA interference significantly reduced the protein expression of COL8A1 in the HGC-27 cell line (Fig 11B, 11D). Colony formation assays showed that knockdown of COL8A1 expression effectively inhibited the colony formation of HGC-27 cells (Fig 11E, 11F). Moreover, Transwell assays indicated that knockdown of COL8A1 expression significantly suppressed the invasion ability of HGC-27 cells (Fig 11G, 11H). CCK-8 assays showed that, compared with the control group, COL8A1 interference significantly inhibited the viability of HGC-27 cells (Fig 11I). These findings suggest that silencing COL8A1 markedly suppresses STAD cell proliferation and migration, supporting its potential role as a pro-tumorigenic factor in STAD progression.

## Discussion

Given the clinical heterogeneity of STAD, effective risk stratification is essential for guiding treatment decisions and improving patient outcomes. Classical pathways, such as those involving caspase and p53, are frequently implicated in the development of drug resistance in gastric cancer cells, complicating effective treatment [32–34]. In contrast,

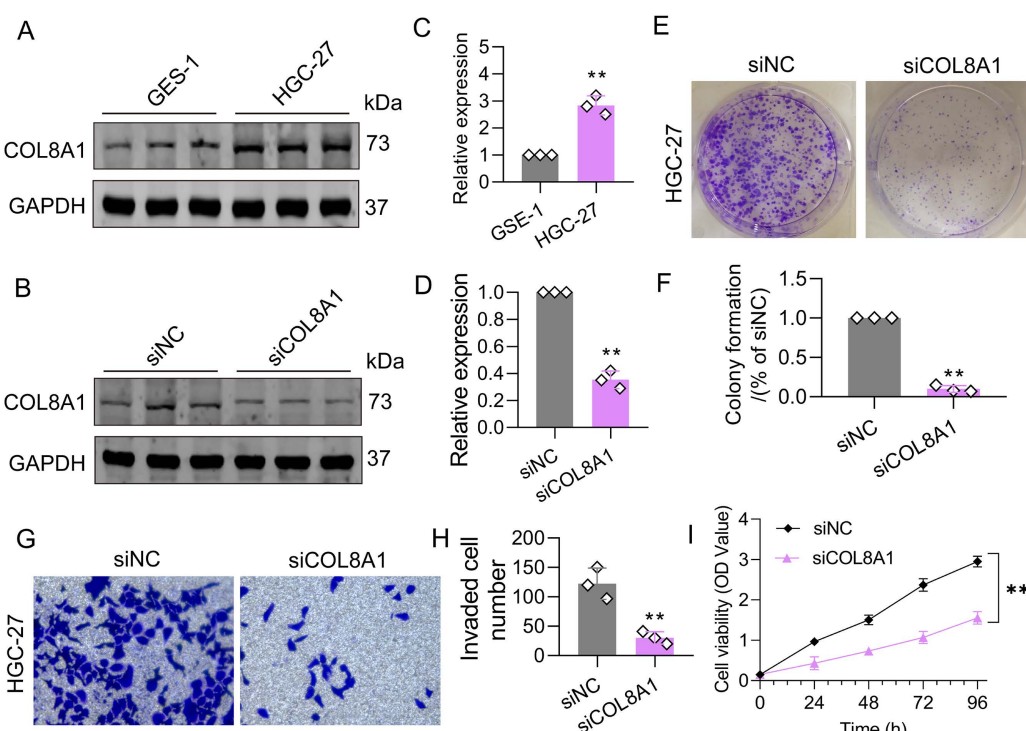

**Fig 11. Suppression of COL8A1 expression significantly reduces the proliferation and migration of STAD cells.** (A, C) WB analysis of COL8A1 expression in GES-1 and HGC-27 cell lines. (B, D) WB analysis of COL8A1 knockdown efficiency in siNC and siCOL8A1 groups. (E, F) Colony formation assays evaluating proliferative capacity. (G, H) Cell invasion assays assessing migratory ability. (I) CCK-8 assay measuring cell viability. *$p < 0.05$; **$p < 0.01$; ***$p < 0.001$; n = 3.

parthanatos is a form of programmed cell death that functions independently of both caspase and p53, offering a promising alternative strategy to overcome treatment resistance in gastric cancer. Moreover, parthanatos is marked by extensive DNA fragmentation and the release of intracellular contents, including damage-associated molecular patterns (DAMPs), which may elicit antitumor immune responses [35]. This suggests that parthanatos may represent a novel form of a novel mechanism of immunogenic cell death, potentially synergizing with immune checkpoint inhibitors to enhance the efficacy of immunotherapy. Therefore, parthanatos-related pathways may possess unique therapeutic potential for the treatment of STAD. Although preliminary studies have suggested a possible link between parthanatos and STAD, no *in vitro* experiments have yet explored the biological effects of parthanatos-related targets on STAD cell behavior [36,37]. In this study, we provide the first *in vitro* evidence demonstrating that modulation of parthanatos-related targets affects the proliferation and migration of STAD cells, offering further support for the functional relevance of parthanatos in gastric cancer.

Parthanatos, along with other regulated forms of cell death such as necroptosis, pyroptosis, and ferroptosis, represents a key hallmark of tumorigenesis and may lead to the development of distinct potential therapeutic strategies [38,39]. Therefore, analyzing the key pathways involved in parthanatos holds clinical significance. Although there are no reports of parthanatos or PRG in STAD, evidence suggests that its key regulator, PARP-1, plays a role in gastric cancer progression and drug resistance. Helicobacter pylori infection has been shown to activate PARP-1, suggesting its involvement in gastric carcinogenesis [40]. Genetic variations in PARP-1 have also been associated with an increased risk of gastric cancer [41], further supporting its potential as a biomarker for the disease [42]. In terms of therapy resistance, PARP-1 contributes to chemotherapy resistance in gastric cancer. It has been demonstrated that inhibiting PARP-1 activity enhances the chemosensitivity of cisplatin-resistant gastric cancer cells [43]. Additionally, TOPBP1 and METTL3 promote resistance to oxaliplatin by upregulating PARP-1 transcription and stabilizing PARP-1 mRNA, respectively, further highlighting its role in drug resistance [44,45]. Targeting PARP-1 has shown promise in overcoming resistance, as the combination of the DNA repair inhibitor ailanthone with a PARP-1 inhibitor synergistically suppresses tumor growth in gastric cancer [46]. These findings suggest that PARP-1, as well as parthanatos process, plays a multifaceted role in gastric cancer, influencing tumorigenesis, chemotherapy resistance, and therapeutic response, making it a potential biomarker and therapeutic target for improving treatment outcomes.

Based on unsupervised consensus clustering of PRG expression profiles, we identified three distinct molecular subtypes, each exhibiting unique immune features and clinical relevance. Subtype A showed moderate immune activity and favorable prognosis, while subtype B was characterized by an immunosuppressive microenvironment, including downregulation of key immune pathways and reduced immune cell infiltration [47]. Interestingly, despite its immunologically "cold" phenotype, subtype B exhibited a clinical prognosis comparable to subtype A, suggesting that immune suppression alone does not necessarily indicate worse outcomes [48]. In contrast, subtype C showed the poorest clinical prognosis, accompanied by upregulation of tumor-promoting pathways and adverse prognostic genes such as COL8A1. Although subtype C exhibited higher immune pathway enrichment, it also showed high TIDE scores and low immunophenoscores, indicating a dysfunctional immune state associated with immune evasion and poor immunotherapeutic response [49,50]. In addition, subtyping based on scRNA-seq data may also partially explain the differences in immunotherapy efficacy [51–53]. Collectively, these findings suggest that molecular subtyping captures clinically meaningful heterogeneity in the tumor immune landscape and offers valuable insights into prognosis and potential immunotherapeutic responsiveness [54].

Our pathway enrichment results show that extracellular matrix-related pathways are enriched across different subgroups, suggesting that the extracellular matrix (ECM) may be a contributing factor to the prognostic differences among the three subtypes. The remodeling of the extracellular matrix (ECM) in tumor tissues can regulate cell interactions by triggering biochemical signals, thereby participating in processes such as proliferation, metastasis, angiogenesis and immune evasion, ultimately influencing cancer progression, metastasis, and dormancy [55,56]. The physical modulation of the ECM, leading to changes in tumor stiffness, also affects drug perfusion and intratumoral distribution, thereby influencing therapeutic efficacy [57]. Therefore, targeting the properties and functions of the ECM as a potential strategy

for anti-malignant treatment has a solid theoretical foundation [58]. Although comprehensive reports are lacking, some evidence suggests the role of the ECM in gastric cancer, including STAD. ECM genes that are significantly upregulated in gastric cancer patients are associated with poor prognosis [59]. Increased ECM density disrupts the E-cadherin/β-catenin complex in gastric cancer cells, thereby modulating GC proliferation and chemotherapy response [60]. However, the relationship between ECM and parthanatos, as well as PARP-1, remains unclear. Our pathway enrichment results suggest a potential connection between them.

In the tumor immune microenvironment of STAD patients with a high PRG score and poorer prognosis, we observed significantly lower levels of CD4+ T cells and neutrophils, as well as significantly higher levels of regulatory T cells compared to the other group, suggesting the presence of an immunosuppressive microenvironment in patients with poor prognosis. We also found higher infiltration levels of mast cells (MCs) in patients with high PRG scores. In solid tumors, depending on tumor type and stage, MC infiltration can either promote or inhibit tumor growth, making them one of the most controversial immune cell types in cancer [61]. In gastric cancer, MCs have been reported to be associated with angiogenesis and metastasis of cancer cells and to promote immunosuppression via the TNFα-PD-L1 pathway [62–65]. This makes MCs a marker of poor prognosis in gastric cancer. In mouse models of gastric cancer, MC activation has also been shown to promote tumor progression by recruiting macrophages [66]. Our findings similarly suggest a tumor-promoting role of MCs in STAD. Although our results suggest the potential value of MCs in risk stratification of STAD, their exact role remains unclear. Further studies are needed to elucidate their biological significance.

Our in vitro experiments demonstrated that COL8A1 promotes the proliferation and invasion of human gastric cancer cell lines. COL8A1 has been reported to enhance metastasis in nasopharyngeal carcinoma by inducing epithelial-mesenchymal transition (EMT) and angiogenesis [67]. The PI3K/AKT signaling pathway plays a key role in COL8A1-mediated EMT and tumor cell proliferation [68,69]. In gastric cancer, upregulation of COL8A1 promotes cell proliferation and indicates poor prognosis [70]. Furthermore, evidence suggests that this prognostic effect is associated with the EMT process [71]. Given the critical role of the PI3K/AKT pathway and EMT in gastric cancer [72,73], targeting COL8A1 in STAD patients is theoretically justified. Additionally, there is evidence indicating that COL8A1, as a marker of fibroblast activation, is a key factor involved in extracellular matrix (ECM) remodeling [74]. Activation of the AKT signaling pathway via PIP3 pathway can further induce COL8A1 expression, thereby contributing to ECM remodeling [75,76]. Given the critical role of ECM remodeling in shaping the tumor microenvironment, promoting tumor invasion, and driving chemoresistance in gastric cancer, COL8A1 emerges as an intriguing therapeutic target worthy of further investigation [77,78]. Another PRG associated with STAD prognosis is SQSTM1, whose alteration is implicated in autophagy and has been linked to gastric tumorigenesis and cancer progression [79]. Furthermore, evidence suggests that enhanced autophagy marked by increased SQSTM1 expression promotes STAD cell migration and survival [80]. Therefore, the identified PRGs warrant further investigation in the context of STAD.

In conclusion, our study comprehensively elucidates the potential role of PRG in the tumor progression and immune microenvironment of STAD through integrative bioinformatics analyses and experimental validation. These findings not only expand our understanding of the molecular mechanisms underlying STAD but also provide novel insights into potential prognostic biomarkers and therapeutic targets. However, several limitations should be acknowledged. First, although our analyses were based on large-scale public databases and multi-omics data, inherent heterogeneity and potential batch effects across datasets may influence the robustness of the results. Second, the functional roles of key genes were only partially validated in vitro; in vivo experiments and clinical data are needed to further confirm their biological and translational relevance. Moreover, the complexity of the tumor immune microenvironment warrants more comprehensive exploration using single-cell or spatial transcriptomic approaches. Despite these limitations, our findings lay a solid foundation for future mechanistic studies and may contribute to the development of precision medicine strategies for gastric cancer. In addition to the datasets used in the current study, we will actively collect or integrate more up-to-date data resources in future research to further validate and expand the applicability and clinical potential of our PRG-based subtyping system and prognostic scoring model.

## Supporting information

**S1 Table. The parthanatos-related gene signatures.**
(XLSX)

**S2. Raw images.**
(PDF)

## Author contributions

**Data curation:** Xiangxin Wu, Zhenhua Liu.

**Methodology:** Yanping Liu, Bowen Wang.

**Software:** Lianming Cai, Yanping Liu, Bowen Wang.

**Validation:** Tianyi Xia, Lianming Cai.

**Visualization:** Bowen Wang.

**Writing – original draft:** Xiangxin Wu.

**Writing – review & editing:** Tianyi Xia, Zhenhua Liu.

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
