## [Decision Letter · Decision Letter 0]

16 Jun 2025

PONE-D-25-27887Multi-omics Analysis of Parthanatos Related Molecular Subgroup and Prognostic Model Development in Stomach AdenocarcinomaPLOS ONE

Dear Dr. Xia,

Thank you for submitting your manuscript to PLOS ONE. After careful consideration, we feel that it has merit but does not fully meet PLOS ONE’s publication criteria as it currently stands. Therefore, we invite you to submit a revised version of the manuscript that addresses the points raised during the review process.

We look forward to receiving your revised manuscript.

Kind regards,

Zhijie Zhao, Ph.D., M.D.,

Academic Editor

PLOS ONE

Journal Requirements:

“This project is supported by Project of Heilongjiang Provincial Health Commission (No. 20230404080327)”

**Additional Editor Comments:**

Revisions are required based on the comments of all reviewers.

Reviewers' comments:

Reviewer's Responses to Questions

**Comments to the Author**

1. Is the manuscript technically sound, and do the data support the conclusions?

Reviewer #1: Yes

Reviewer #2: Yes

2. Has the statistical analysis been performed appropriately and rigorously? 

Reviewer #1: Yes

Reviewer #2: Yes

3. Have the authors made all data underlying the findings in their manuscript fully available?

Reviewer #1: Yes

Reviewer #2: Yes

4. Is the manuscript presented in an intelligible fashion and written in standard English?

Reviewer #1: Yes

Reviewer #2: No

5. Review Comments to the Author

Reviewer #1: The paper titled "Multi-omics Analysis of Parthanatos Related Molecular Subgroup and Prognostic Model Development in Stomach Adenocarcinoma" conducts a systematic investigation into the influence of parthanatos-associated genes in STAD. By integrating transcriptomic, genomic, and clinical data from the TCGA and GEO databases, the study defines multiple molecular subgroups that exhibit distinct parthanatos-related gene expression profiles. These clusters exhibit differences in immune infiltration status, mutational profiles, and clinical outcomes. Noteworthy is one subgroup that shows significantly elevated AIFM1 and PARP1 expression, immune evasion markers, and poor survival rates, implying a potential association between disrupted parthanatos signaling and enhanced tumor malignancy.

The authors stratify patients into high- and low-risk groups with notable survival differences using integrative analysis to create a reliable prognostic model based on parthanatos gene signatures. This model demonstrates strong predictive performance across multiple cohorts and remains independent of conventional clinical variables in multivariate analyses.The study also assesses therapeutic vulnerabilities by associating gene expression with drug response patterns, which enables the identification of promising agents for precision therapy, particularly in patients with high-risk profiles. However, several limitations exist. Since the analysis depends on retrospective datasets, its clinical applicability remains limited without further prospective confirmation. Additionally, more in-depth functional studies—both in vitro and in vivo—are necessary to clarify the mechanistic involvement of the candidate genes in parthanatos and tumor development. Future research that integrates spatial or single-cell transcriptomic technologies may enhance the understanding of the tumor microenvironment in relation to parthanatos. Collectively, the study offers meaningful insights into how parthanatos influences prognosis and treatment strategies in gastric cancer.

1. What is the current state of research on pyroptosis-associated genes in relation to STAD?

2. What publicly available platforms provide the transcriptomic and clinical information for STAD in this study, and how is this information processed? Including references such as Zn‐DHM Nanozymes Enhance Muscle Regeneration Through ROS Scavenging and Macrophage Polarization in Volumetric Muscle Loss Revealed by Single-Cell Profiling, Single-cell RNA sequencing and immune microenvironment analysis reveal PLOD2-driven malignant transformation in cervical cancer.

3.By what methods are DE-PARGs identified, and how is molecular subtype classification based on these genes implemented?

4. Which machine learning models are applied to create the PARG-based prognostic scoring system? Supporting references can be helpful.

5. What molecular classifications are produced from the PARG-based analysis of STAD, and what correlations exist with clinical outcomes?

6. Immune infiltration varies significantly among different PARG molecular subtypes, which may correlate with varied responses to immunotherapy. This hypothesis is supported by studies such as PMID: 40421026 and 40406148.

7. Each gene subtype linked to PARG classification reveals unique biological characteristics and impacts patient prognosis differently. Supporting literature, such as DOI: 10.15212/bioi-2022-0008 and PMID: 27672669, explores these relationships in more detail.

8. At the same time, substantial variation in tumor mutation burden and immune response is observed across PARG scoring subgroups, suggesting that PARG-based stratification could optimize immunotherapy approaches.

9. Although mast cells are known to be part of the STAD immune landscape, their exact role remains unclear. Consequently, future investigations with strong literature support are required to clarify their biological significance.

Reviewer #2: This study conducts a multi-omics analysis of parthanatos-related genes (PARGs) in stomach adenocarcinoma (STAD), identifying molecular subtypes and constructing a PARG-based prognostic model. The integration of transcriptomic, genomic, single-cell, and functional data is well designed. The topic is relatively novel and shows clinical application potential.

However, the manuscript still has several issues regarding scientific rigor, clarity of logic, and language accuracy. Major revision is recommended before it can be considered for publication. Specific comments are as follows:

1. The manuscript contains a number of expressions influenced by Chinese syntax. A thorough language polish is needed to improve clarity, professionalism, and readability.

2. The current datasets are relatively old. The authors are encouraged to supplement with more recent publicly available datasets or their own cohort to enhance robustness and timeliness.

3. Figure legends should be integrated into the result descriptions as per journal guidelines, rather than being placed as isolated paragraphs.

4. In Figure 2, the functional differences between cluster A and cluster C are not clearly discussed. A more detailed KEGG enrichment comparison between these clusters is needed.

5. The color assignments for clusters A, B and C are not consistent across figures. Please unify the color scheme to maintain visual consistency.

6. This is a highlight of the study but is underdeveloped. The authors are advised to incorporate deeper cell–cell communication analyses (e.g., using CellChat) to better characterize TAM heterogeneity and immune-stromal interactions.

7. Although COL8A1 knockdown was experimentally validated, the underlying mechanism is not sufficiently discussed. Please explore its potential involvement in EMT, PI3K/AKT pathway, or ECM remodeling, and consider validating these axes.

8. The use of “PARG” to represent “parthanatos-related genes” is potentially misleading, as PARG typically refers to poly(ADP-ribose) glycohydrolase. Please revise and unify the terminology throughout the manuscript.

9. Although parthanatos is a relatively novel concept in cancer biology, the authors should further clarify its unique value in gastric cancer and elaborate on how their findings differ from previous studies.

10. Ensure that figure numbering and font sizes are consistent across all figures to improve overall visual presentation.

Also,the references cited in this article are not sufficient, and there is a lack of in-depth comparative discussion. Background and methodology also require further literature support. Some related research should be cited:

1. Role of natural products in tumor therapy from basic research and clinical perspectives, 10.15212/AMM-2023-0050

2. Identifying the key genes of Epstein-Barr virus-regulated tumour immune microenvironment of gastric carcinomas�10.1111/cpr.13373

3. Single-cell RNA sequencing reveals immune cell dysfunction in the peripheral blood of patients with highly aggressive gastric cancer�10.1111/cpr.13591

4. Macrophage differentiation in enhancing hematopoietic function of ribonucleic acid for injection II via multi-omics analysis, 10.15212/AMM-2024-0001

5. Prospective study and validation of early warning marker discovery based on integrating multi-omics analysis in severe burn patients with sepsis, 10.1093/burnst/tkac050

6. PLOS authors have the option to publish the peer review history of their article (what does this mean? ). If published, this will include your full peer review and any attached files.

**Do you want your identity to be public for this peer review?** For information about this choice, including consent withdrawal, please see our Privacy Policy .

Reviewer #1: No

Reviewer #2: No

---

## [Author Response · Author response to Decision Letter 1]

25 Jun 2025

We sincerely appreciate the editor and all reviewers for their valuable comments and suggestions, which have significantly improved the quality and clarity of our manuscript. We have carefully considered each comment and revised the manuscript accordingly. Below, we provide a detailed point-by-point response to each comment, with our responses highlighted in bold and all modifications clearly marked in the revised manuscript.

Review Comments to the Author

Reviewer #1: The paper titled "Multi-omics Analysis of Parthanatos Related Molecular Subgroup and Prognostic Model Development in Stomach Adenocarcinoma" conducts a systematic investigation into the influence of parthanatos-associated genes in STAD. By integrating transcriptomic, genomic, and clinical data from the TCGA and GEO databases, the study defines multiple molecular subgroups that exhibit distinct parthanatos-related gene expression profiles. These clusters exhibit differences in immune infiltration status, mutational profiles, and clinical outcomes. Noteworthy is one subgroup that shows significantly elevated AIFM1 and PARP1 expression, immune evasion markers, and poor survival rates, implying a potential association between disrupted parthanatos signaling and enhanced tumor malignancy.

The authors stratify patients into high- and low-risk groups with notable survival differences using integrative analysis to create a reliable prognostic model based on parthanatos gene signatures. This model demonstrates strong predictive performance across multiple cohorts and remains independent of conventional clinical variables in multivariate analyses.The study also assesses therapeutic vulnerabilities by associating gene expression with drug response patterns, which enables the identification of promising agents for precision therapy, particularly in patients with high-risk profiles. However, several limitations exist. Since the analysis depends on retrospective datasets, its clinical applicability remains limited without further prospective confirmation. Additionally, more in-depth functional studies—both in vitro and in vivo—are necessary to clarify the mechanistic involvement of the candidate genes in parthanatos and tumor development. Future research that integrates spatial or single-cell transcriptomic technologies may enhance the understanding of the tumor microenvironment in relation to parthanatos. Collectively, the study offers meaningful insights into how parthanatos influences prognosis and treatment strategies in gastric cancer.

1. What is the current state of research on pyroptosis-associated genes in relation to STAD?

Reply:

We sincerely thank the reviewer for their insightful comments and fully understand your interest in the recent advances regarding programmed cell death mechanisms in STAD. It is important to clarify that the focus of our study is not on pyroptosis, but rather on a systematic investigation of parthanatos-related genes (PRG) in STAD, including their expression patterns, biological functions, and prognostic value.

PRG is a distinct form of programmed cell death triggered by hyperactivation of PARP1, independent of apoptosis and pyroptosis. It plays a unique role in tumor metabolic dysregulation, DNA damage response, and immune microenvironment modulation. We are aware that recent studies have suggested potential crosstalk and regulatory interactions among various forms of cell death, such as parthanatos, pyroptosis, and ferroptosis. These findings offer promising directions for future research into tumor heterogeneity.

In the revised manuscript, we have further clarified that our study focuses specifically on parthanatos and have incorporated a brief discussion in both the Introduction and Discussion sections regarding the potential interplay between parthanatos and other forms of programmed cell death, such as pyroptosis. This aims to help readers better understand the context and significance of our work.

We sincerely appreciate your attention to this important area of research, which has provided us with valuable insights and potential avenues for future exploration.

Line 78-line 81; Line 664-667

2. What publicly available platforms provide the transcriptomic and clinical information for STAD in this study, and how is this information processed? Including references such as Zn‐DHM Nanozymes Enhance Muscle Regeneration Through ROS Scavenging and Macrophage Polarization in Volumetric Muscle Loss Revealed by Single-Cell Profiling, Single-cell RNA sequencing and immune microenvironment analysis reveal PLOD2-driven malignant transformation in cervical cancer.

Reply:

Thank you for your valuable comments. In this study, the transcriptomic expression matrices and corresponding clinical baseline information of NC tissues and STAD tissues were obtained from two publicly accessible databases: The Cancer Genome Atlas (TCGA) and the Gene Expression Omnibus (GEO). The specific data sources and processing steps are as follows:

1: TCGA database (https://portal.gdc.cancer.gov/):

Transcriptomic expression data (in count format) and clinical information of STAD samples were downloaded using the R programming environment. Gene annotation and matrix processing were performed using human genome annotation files and handled via Perl scripts. Samples lacking survival information or with an overall survival (OS) of less than 30 days were excluded. After filtering, 337 STAD samples and 32 normal gastric tissue samples were included for subsequent analyses.

2: GEO database (https://www.ncbi.nlm.nih.gov/geo/):

Two STAD-related gene expression datasets were included: GSE84437 and GSE15459.

(1) GSE84437 (platform: GPL6947, Illumina HumanHT-12 V3.0) contains 431 STAD samples.

(2) GSE15459 (platform: GPL570, Affymetrix HG-U133 Plus 2.0) includes 182 STAD samples.

The original data were annotated and normalized using the corresponding platform annotation files via Perl scripts. Consistent with the TCGA cohort, samples lacking survival data or with OS < 30 days were excluded.

Data integration and normalization:

For the TCGA-STAD dataset, count data were converted into TPM (Transcripts Per Kilobase Million) format to enhance cross-sample comparability, followed by log2(TPM+1) normalization. Subsequently, TCGA and GSE84437 datasets were merged to construct the training cohort. Batch effects and normalization were corrected using the “sva” and “limma” packages in R. The GSE15459 dataset was used as an independent external validation cohort.

We have further optimized the manuscript content and cited the recommended references to improve its readability and scientific rigor.

Line 92-line 106; line 111-line 114

3.By what methods are DE-PRGs identified, and how is molecular subtype classification based on these genes implemented?

Reply:

We sincerely thank the reviewer for raising this insightful and professional question. The identification of differentially expressed parthanatos-related genes (DE-PRGs) and the classification process of molecular subtypes in our study were conducted as follows:

First, differential expression analysis was performed using transcriptomic data from tumor and normal tissues in the TCGA-STAD cohort, processed with the “limma” R package. The filtering criteria were set as |log2 fold change (FC)| ≥ 2 and adjusted P-value < 0.05. Genes meeting these thresholds and present in our predefined list of parthanatos-related genes (PRGs) were defined as DE-PRGs.

Next, unsupervised consensus clustering based on the expression profiles of these DE-PRGs was carried out to identify potential molecular subtypes of STAD. This was implemented using the “ConsensusClusterPlus” R package with the partitioning around medoids (PAM) algorithm. The clustering was repeated 1,000 times to ensure robustness, and we evaluated the CDF plots for cluster numbers (k) ranging from 2 to 9. The optimal number of clusters was determined based on the delta area plot and clustering stability.

To further validate the separation of identified subtypes, PCA was performed for dimensionality reduction and visualization. In addition, we compared the immune infiltration profiles and prognostic differences among the subtypes to explore their biological significance.

We sincerely appreciate your attention to the methodological details, which greatly enhances the transparency and reproducibility of our study.

Line 119-line 122; line 125-line 132

4. Which machine learning models are applied to create the PRG-based prognostic scoring system? Supporting references can be helpful.

Reply:

We sincerely thank the reviewer for the valuable comments. To enhance the robustness and accuracy of the prognostic scoring system based on PRGs, we systematically evaluated and compared 10 commonly used machine learning algorithms, including: CoxBoost, Enet, GBM, Lasso, plsRcox, Ridge, RSF, Stepwise Cox regression, SuperPC, and survival-SVM.

Model construction and evaluation were conducted using the corresponding R packages: “survival,” “survivalsvm,” “glmnet,” “randomForestSRC,” “CoxBoost,” “gbm,” and “Superpc.” The predictive performance of each model was assessed using the concordance index (C-index), and model stability was evaluated via 10-fold cross-validation. Based on these comparisons, the model with the highest C-index and strong generalization performance was selected to construct the final PRG-based prognostic scoring system. This model demonstrated favorable survival prediction performance in both the training cohort and external validation cohort, further supporting its clinical applicability.

In response to the reviewer’s suggestion, we have now provided a detailed description of all machine learning algorithms, analysis workflows, and performance evaluation metrics in the “Materials and Methods” section. Additionally, relevant supporting references have been added. We truly appreciate your attention to the modeling methodology, which has greatly helped us improve the clarity and completeness of this section.

Line 180-line 185

5. What molecular classifications are produced from the PRG-based analysis of STAD, and what correlations exist with clinical outcomes?

Reply:

We sincerely thank the reviewer for raising this important question. In our study, we performed unsupervised consensus clustering analysis based on the expression profiles of DE-PRG to classify STAD samples into three distinct molecular subtypes: Cluster A, Cluster B, and Cluster C. The clustering process was conducted using the “ConsensusClusterPlus” R package. The CDF curves and delta area plots were used to evaluate clustering stability across different values of k, and k = 3 was determined to be the optimal number of clusters.

To explore the clinical relevance of these subtypes, we conducted the following analyses:

1: Survival analysis: Kaplan-Meier survival curves showed that patients in Cluster B had significantly better OS compared to those in Cluster A and Cluster C (P < 0.001), suggesting that the subtype classification provides clear prognostic stratification.

2: Tumor immune microenvironment analysis: Using ssGSEA and the ESTIMATE algorithm, we assessed the infiltration levels of immune-related cells within the tumor microenvironment. We observed that Cluster B exhibited lower levels of immune cell infiltration, which may be associated with its favorable clinical outcomes.

We greatly appreciate your insightful suggestion. We believe that the further clarification of the molecular subtypes and their clinical correlations substantially enhances the scientific significance and translational potential of our study.

6. Immune infiltration varies significantly among different PRG molecular subtypes, which may correlate with varied responses to immunotherapy. This hypothesis is supported by studies such as PMID: 40421026 and 40406148.

Reply

We sincerely thank the reviewer for this important comment. We fully agree that the distinct immune infiltration profiles observed among different PRG molecular subtypes may influence their responsiveness to immunotherapy. The results of our immune characterization analyses support this hypothesis: we applied multiple algorithms, including ssGSEA and ESTIMATE, to assess immune infiltration across the subtypes. Our findings revealed the following patterns:

1: Cluster A exhibited a moderate level of immune activation.

2: Cluster B showed significantly lower immune cell infiltration, indicating an “immune-cold” tumor phenotype.

3: Cluster C, despite demonstrating a high level of immune infiltration, also displayed elevated TIDE scores and reduced IPS scores, suggesting a state of immune dysfunction and immune escape.

These results further support the notion that PRG-based molecular subtypes may be closely associated with differential immunotherapy responses.

In accordance with your suggestion, we have incorporated this discussion into the revised “Discussion” section and have cited the two relevant references to strengthen the supporting evidence. We believe that these additions further enhance the scientific value of our study, particularly in the context of guiding immunotherapy strategies.

Line 695-line 702

7. Each gene subtype linked to PRG classification reveals unique biological characteristics and impacts patient prognosis differently. Supporting literature, such as DOI: 10.15212/bioi-2022-0008 and PMID: 27672669, explores these relationships in more detail.

Reply:

We sincerely appreciate your valuable suggestion. As requested, we have now provided a more explicit explanation regarding the correlation between PRG molecular subtyping and patient prognosis, supported by relevant literature citations. Thank you for your insightful comment, which has helped improve the clarity of our study.

Line 685-line 695

8. At the same time, substantial variation in tumor mutation burden and immune response is observed across PRG scoring subgroups, suggesting that PRG-based stratification could optimize immunotherapy approaches.

Reply:

We sincerely thank the reviewer for the insightful comments. Indeed, in this study, we conducted TMB analysis and immune response evaluation between the high and low PRG score groups. The results are as follows:

1: TMB analysis revealed that the low PRG score group exhibited a significantly higher TMB level, which is consistent with greater immunogenic potential.

2: Immune response prediction, as assessed by Immunophenoscore (IPS), showed that the low PRG score group had a higher predicted immunotherapy response potential, suggesting that these patients may be more likely to benefit from immune checkpoint inhibitor (ICI) treatment.

These findings support your perspective that the PRG score not only serves as a prognostic indicator but also holds promise as a predictor of immunotherapy responsiveness. The PRG-based stratification strategy may thus serve as a valuable tool for optimizing patient selection and tailoring personalized immunotherapeutic regimens in STAD.

Following your suggestion, we have incorporated the relevant comparative analyses into the “Discussion” section and further emphasized the potential clinical utility of the PRG scoring system in guiding immunotherapy decisions.

Once again, we deeply appreciate your insightful and constructive suggestions, which have greatly helped us to better articulate the link between PRG scoring and tumor immunity, and to strengthen the translational relevance of our study.

9. Although mast cells are known to be part of the STAD immune landscape, their exact role remains unclear. Consequently, future investigations with strong literature support are required to clarify their biological significance.

Reply:

We sincerely appreciate the reviewer’s forward-looking and constructive suggestion. Indeed, mast cells, as important components of the tumor immune microenvironment, have been observed to infiltrate STAD tissues to a certain extent. However, their precise role in tumor initiation and progression remains inconclusive, and no consensus has been reached in the current literature.

In our study, we evaluated immune cell infiltration using the ssGSEA algorithm, which included an analysis of mast cell activation status across different PRG-defined subtypes. The resul

---

## [Decision Letter · Decision Letter 1]

5 Aug 2025

PONE-D-25-27887R1

Multi-omics Analysis of Parthanatos Related Molecular Subgroup and Prognostic Model Development in Stomach Adenocarcinoma

PLOS ONE

Dear Dr. Xia,

Thank you for submitting your manuscript to PLOS ONE. After careful consideration, we feel that it has merit but does not fully meet PLOS ONE’s publication criteria as it currently stands. Therefore, we invite you to submit a revised version of the manuscript that addresses the points raised during the review process.

 Note from the Editorial Office: I believe you have added a box outline around the wrong set of bands in the original blot image for COL8A1 in Fig. 11B. Please correct this, then please combine the original blot images into a single file, which should be uploaded as a PDF file named "S1_raw_images" and uploaded as a Supporting Information File, per our submission guidelines: https://journals.plos.org/plosbiology/s/figures#loc-blot-and-gel-reporting-requirements Thank you for your attention to these requests.

We look forward to receiving your revised manuscript.

Kind regards,

Jin Su Kim, PhD

Academic Editor

PLOS ONE

Journal Requirements:

Reviewers' comments:

Reviewer's Responses to Questions

**Comments to the Author**

1. If the authors have adequately addressed your comments raised in a previous round of review and you feel that this manuscript is now acceptable for publication, you may indicate that here to bypass the “Comments to the Author” section, enter your conflict of interest statement in the “Confidential to Editor” section, and submit your "Accept" recommendation.

Reviewer #1: All comments have been addressed

Reviewer #3: All comments have been addressed

Reviewer #4: (No Response)

2. Is the manuscript technically sound, and do the data support the conclusions?

Reviewer #1: Yes

Reviewer #3: Yes

Reviewer #4: Yes

3. Has the statistical analysis been performed appropriately and rigorously? 

Reviewer #1: Yes

Reviewer #3: Yes

Reviewer #4: Yes

4. Have the authors made all data underlying the findings in their manuscript fully available?

Reviewer #1: Yes

Reviewer #3: Yes

Reviewer #4: Yes

5. Is the manuscript presented in an intelligible fashion and written in standard English?

Reviewer #1: Yes

Reviewer #3: Yes

Reviewer #4: Yes

6. Review Comments to the Author

Reviewer #1: (No Response)

Reviewer #3: The author has made detailed revisions to the comments of the previous reviewer, thus resolving many issues of imprecise expression. However, after reading the manuscript, the only thing I feel inappropriate is that the author's conclusion is too lengthy. The conclusion should be a highly condensed and summarized version of the research findings, rather than a secondary repetition of the entire research process. It is recommended that the author streamline the conclusion.

Reviewer #4: This study, for the first time through integrated multi-omics analysis, systematically elaborates on the molecular subtypes, prognostic value, and immune microenvironment regulatory mechanisms of parthanatos-related genes (PRGs) in stomach adenocarcinoma (STAD), and experimentally verifies the cancer-promoting function of the key gene COL8A1. The research design is rigorous with advanced methods (covering transcriptome, single-cell sequencing, machine learning, etc.) and has clear potential for clinical translation. The manuscript has been substantially revised according to the reviewers' comments, but there are still some issues that need further improvement.

Main Advantages

Innovation and Clinical Significance

(1) For the first time, a molecular typing system (Cluster A/B/C) of PRGs in STAD is established, revealing its associations with prognosis, immune infiltration, and treatment response, which provides a new perspective for precision therapy.

(2) The PRG prognostic model (StepCox + Enet) constructed based on 10 machine learning algorithms shows robustness in both the training set and the validation set (GSE15459) (with outstanding C-index) and is independent of clinicopathological factors.

(3) Experimental verification of the function of COL8A1 in promoting the proliferation/migration of STAD cells supports its potential as a therapeutic target.

Methodological Rigor

(1) The integration of multi-omics data is comprehensive (TCGA, GEO cohorts) with strict quality control (excluding samples with OS < 30 days).

(2) Single-cell analysis (GSE163558) combined with CellChat reveals the cell interaction network in the tumor microenvironment, enhancing the reliability of the conclusions.

(3) The immune microenvironment assessment is diversified (ESTIMATE, ssGSEA, TIDE, IPS), corroborating the association between PRG subtypes and immune therapy response.

Key Issues and Improvement Suggestions

Insufficient Depth of Molecular Mechanisms (Key Defect)

Issue: The cancer-promoting mechanism of COL8A1 only stays at the phenotypic level (proliferation/migration), lacking verification at the pathway level (such as EMT, PI3K/AKT). The discussion mentions that "PI3K/AKT induces COL8A1-mediated ECM remodeling" (Line 748), but there is no experimental support.

Suggestion: Supplement WB verification of EMT markers (E-cadherin/Vimentin) or PI3K/AKT pathway proteins after COL8A1 knockout; or clearly list mechanism research as a future direction and emphasize this limitation in the discussion.

Contradictions in Immune Microenvironment Analysis

Issue: Cluster C shows high immune infiltration but the worst prognosis, and the authors attribute it to "immune dysfunction" (Line 696-698), but there is a lack of key evidence (such as T cell exhaustion markers, proportion of immunosuppressive cells).

Suggestion: Supplement the expression heatmap of exhaustion markers such as PD-1/CTLA-4 and LAG-3 in Cluster C. Analyze the differences in infiltration of Treg/MDSC between high and low PRG score groups (the existing ssGSEA only provides an overview of 23 cell types).

7. PLOS authors have the option to publish the peer review history of their article (what does this mean? ). If published, this will include your full peer review and any attached files.

**Do you want your identity to be public for this peer review?** For information about this choice, including consent withdrawal, please see our Privacy Policy .

Reviewer #1: No

Reviewer #3: No

Reviewer #4: No

---

## [Author Response · Author response to Decision Letter 2]

7 Aug 2025

We sincerely appreciate the editor and all reviewers for their valuable comments and suggestions, which have significantly improved the quality and clarity of our manuscript. We have carefully considered each comment and revised the manuscript accordingly. Below, we provide a detailed point-by-point response to each comment, with our responses highlighted in bold and all modifications clearly marked in the revised manuscript.

Review Comments to the Author

Reviewer #1: (No Response)

Reviewer #3: The author has made detailed revisions to the comments of the previous reviewer, thus resolving many issues of imprecise expression. However, after reading the manuscript, the only thing I feel inappropriate is that the author's conclusion is too lengthy. The conclusion should be a highly condensed and summarized version of the research findings, rather than a secondary repetition of the entire research process. It is recommended that the author streamline the conclusion.

Reply

We sincerely thank you for taking the time to review our manuscript and for providing your valuable feedback. We highly appreciate your comment regarding the "overly lengthy conclusion section." Upon careful review, we agree that the original conclusion was not sufficiently concise and partially repeated content from earlier sections of the manuscript.

In accordance with your suggestion, we have revised and streamlined the conclusion to focus more clearly on the core findings and key innovations of our study. Redundant descriptions of the research process have been removed to enhance clarity, coherence, and summarization.

Once again, we are truly grateful for your insightful comments and recognition of our work. Your suggestions have significantly contributed to improving the quality of the manuscript, and we sincerely hope that the revised version meets your expectations.

Line 78-Line 85; line 773-line 778

Reviewer #4: This study, for the first time through integrated multi-omics analysis, systematically elaborates on the molecular subtypes, prognostic value, and immune microenvironment regulatory mechanisms of parthanatos-related genes (PRGs) in stomach adenocarcinoma (STAD), and experimentally verifies the cancer-promoting function of the key gene COL8A1. The research design is rigorous with advanced methods (covering transcriptome, single-cell sequencing, machine learning, etc.) and has clear potential for clinical translation. The manuscript has been substantially revised according to the reviewers' comments, but there are still some issues that need further improvement.

Main Advantages

Innovation and Clinical Significance

(1) For the first time, a molecular typing system (Cluster A/B/C) of PRGs in STAD is established, revealing its associations with prognosis, immune infiltration, and treatment response, which provides a new perspective for precision therapy.

(2) The PRG prognostic model (StepCox + Enet) constructed based on 10 machine learning algorithms shows robustness in both the training set and the validation set (GSE15459) (with outstanding C-index) and is independent of clinicopathological factors.

(3) Experimental verification of the function of COL8A1 in promoting the proliferation/migration of STAD cells supports its potential as a therapeutic target.

Methodological Rigor

(1) The integration of multi-omics data is comprehensive (TCGA, GEO cohorts) with strict quality control (excluding samples with OS < 30 days).

(2) Single-cell analysis (GSE163558) combined with CellChat reveals the cell interaction network in the tumor microenvironment, enhancing the reliability of the conclusions.

(3) The immune microenvironment assessment is diversified (ESTIMATE, ssGSEA, TIDE, IPS), corroborating the association between PRG subtypes and immune therapy response.

Key Issues and Improvement Suggestions

Insufficient Depth of Molecular Mechanisms (Key Defect)

Issue: The cancer-promoting mechanism of COL8A1 only stays at the phenotypic level (proliferation/migration), lacking verification at the pathway level (such as EMT, PI3K/AKT). The discussion mentions that "PI3K/AKT induces COL8A1-mediated ECM remodeling" (Line 748), but there is no experimental support.

Suggestion: Supplement WB verification of EMT markers (E-cadherin/Vimentin) or PI3K/AKT pathway proteins after COL8A1 knockout; or clearly list mechanism research as a future direction and emphasize this limitation in the discussion.

Reply

Thank you for your thoughtful review and valuable suggestions regarding our study. You rightly pointed out that the current manuscript lacks in-depth mechanistic validation of how COL8A1 promotes tumor progression, particularly in terms of experimental evidence involving EMT markers (such as E-cadherin and Vimentin) and the PI3K/AKT signaling pathway.

We fully agree with your insightful comments. Due to experimental limitations and manuscript length constraints, we were unable to include additional Western blot analyses in the current version. However, we have clearly acknowledged this limitation in the Discussion section and have elaborated on our plan to conduct further mechanistic investigations—specifically, to validate the involvement of EMT processes and the PI3K/AKT pathway—in future studies.

We believe that these additions will enhance the scientific rigor and logical coherence of the manuscript. Once again, we sincerely appreciate your constructive feedback and recognition of our work.

Line 760-Line 772

Contradictions in Immune Microenvironment Analysis

Issue: Cluster C shows high immune infiltration but the worst prognosis, and the authors attribute it to "immune dysfunction" (Line 696-698), but there is a lack of key evidence (such as T cell exhaustion markers, proportion of immunosuppressive cells).

Suggestion: Supplement the expression heatmap of exhaustion markers such as PD-1/CTLA-4 and LAG-3 in Cluster C. Analyze the differences in infiltration of Treg/MDSC between high and low PRG score groups (the existing ssGSEA only provides an overview of 23 cell types).

Reply

We sincerely thank the reviewer for the professional and constructive comments. In response to the concern regarding the lack of sufficient evidence in the immune microenvironment analysis, we have further supplemented the expression profiles of T cell exhaustion markers, including PD-1, CTLA-4, and LAG-3, across different subtypes (including Cluster C). These results have been incorporated into the revised manuscript.

Additionally, we have analyzed and presented the infiltration levels of immunosuppressive cells such as regulatory T cells (Tregs) and MDSCs in the high and low PRG score groups, as shown in Figure 8E. The relevant differences are also highlighted in the figure.

We believe that these additions provide stronger support for the potential association between "immune dysfunction" and poor prognosis. Once again, we truly appreciate your insightful guidance on our work.

Line 405-Line 410; line 421-line 422, Line 540-line 544, Figure 3J-M, Figure 8E

---

## [Editor Report · Decision Letter 2]

9 Sep 2025

Multi-omics Analysis of Parthanatos Related Molecular Subgroup and Prognostic Model Development in Stomach Adenocarcinoma

PONE-D-25-27887R2

Dear Dr. Tianyi Xia

We’re pleased to inform you that your manuscript has been judged scientifically suitable for publication and will be formally accepted for publication once it meets all outstanding technical requirements.

Kind regards,

Jin Su Kim, PhD

Academic Editor

PLOS ONE
---

## [Editor Report · Acceptance letter]

PONE-D-25-27887R2

PLOS ONE

Dear Dr. Xia,

I'm pleased to inform you that your manuscript has been deemed suitable for publication in PLOS ONE. Congratulations! Your manuscript is now being handed over to our production team.

Kind regards,

on behalf of

Dr. Jin Su Kim

Academic Editor

PLOS ONE